# Self acceleration from spectral geometry in dissipative quantum-walk dynamics

Peng Xue [1] ✉, Quan Lin[1], Kunkun Wang[2], Lei Xiao[1], Stefano Longhi [3,4] ✉ & Wei Yi [5,6] ✉

The dynamic behavior of a physical system often originates from its spectral properties. In open systems, where the effective non-Hermitian description enables a wealth of spectral structures in the complex plane, the concomitant dynamics are significantly enriched, whereas the identification and comprehension of the underlying connections are challenging. Here we experimentally demonstrate the correspondence between the transient self-acceleration of local excitations and the non-Hermitian spectral topology using lossy photonic quantum walks. Focusing first on one-dimensional quantum walks, we show that the measured short-time acceleration of the wave function is proportional to the area enclosed by the eigenspectrum. We then reveal a similar correspondence in two-dimension quantum walks, where the self-acceleration is proportional to the volume enclosed by the eigenspectrum in the complex parameter space. In both dimensions, the transient self-acceleration crosses over to a long-time behavior dominated by a constant flow at the drift velocity. Our results unveil the universal correspondence between spectral topology and transient dynamics, and offer a sensitive probe for phenomena in non-Hermitian systems that originate from spectral geometry.

In both classical and quantum mechanics, the dynamics of a system are intimately connected with its spectral features through the equations of motion[1-4]. Just as the energy of a celestial body impacts its trajectory[5], the energy quantization accounts for the spontaneous collapse and revival in quantum models[2]. In solid materials, transport of electrons depends on the lattice dispersion[6-8], with strict connections between spectral and dynamical features of transport. For example, in a clean lattice with absolutely continuous spectrum, transport is ballistic, while in disordered lattices, the different nature of the energy spectrum greatly impacts the spreading dynamics of the wave function, leading to distinct behaviors ranging from the Anderson localization to diffusive and intermittent quantum dynamics[9]. These examples, however, all concern isolated systems with completely real energy spectra. For open systems that exchange energy or particles with its environment, an effective non-Hermitian description is often adopted, where the underpinning non-Hermitian Hamiltonians feature complex eigenspectra[10-12]. This enables a rich variety of spectral geometries in the complex plane, with non-trivial consequences on the system behavior[13-21]. The dynamics generated by non-Hermitian Hamiltonians are often less intuitive than those of conventional Hermitian systems. For example, the semiclassical equations of non-Hermitian Hamiltonians generalize the Ehrenfest theorem in a nontrivial way[22], leading to phase-space dynamics with a changing metric structure[22-24]. Beyond the semiclassical models, recent studies of non-Hermitian Hamiltonians with the non-Hermitian skin effect[17,25] unravelled that, spectral features such as the closing of

[1]School of Physics, Southeast University, Nanjing 211189, China. [2]School of Physics and Optoelectronic Engineering, Anhui University, Hefei 230601, China. [3]Dipartimento di Fisica, Politecnico di Milano, Piazza L. da Vinci 32, Milano I-20133, Italy. [4]IFISC (UIB-CSIC) Instituto de Fisica Interdisciplinar y Sistemas Complejos, Palma de Mallorca E-07122, Spain. [5]CAS Key Laboratory of Quantum Information, University of Science and Technology of China, Hefei 230026, China. [6]CAS Center For Excellence in Quantum Information and Quantum Physics, Hefei 230026, China. ✉e-mail: gnep.eux@gmail.com; stefano.longhi@polimi.it; wyiz@ustc.edu.cn

the imaginary gap on the complex plane[25–32], or the overall spectral topology[33–41], can have detectable dynamic consequences, including anomalous relaxation dynamics[42–44], boundary accumulations of loss in the dynamics (known as the edge burst)[30–32], and the persistent directional flow which has served as an experimental signature for the non-Hermitian skin effect[18,19,45–55]. However, as most of such dynamic behaviors only dominate at long times and require post selection to avoid quantum jumps, their experimental identification in genuinely quantum systems may be challenging.

In this work, we experimentally reveal the impact of non-Hermitian spectral topology in transient dynamics, by studying the propagation of a local excitation along a dissipative lattice using photonic quantum walks. We show that the short-time, center-of-mass acceleration of the wave function, dubbed self-acceleration[56] because of the absence of any external force, is proportional to the area enclosed by the eigenenergy spectrum of the system on the complex plane. While the direction of the propagation is given by the spectral winding number, the self-acceleration vanishes at long times, giving way to a directional flow with a constant drift velocity. The correspondence between the spectral geometry and bulk dynamics also persists in a wide class of two-dimensional systems, for which we demonstrate that the self-acceleration becomes proportional to the volume enclosed by the eigenspectrum in the complex parameter space. Our experiment establishes a fundamental correspondence between the spectral geometry and short-time dynamics in non-Hermitian systems, complementing existing experiments on the long-time dynamics and chiral amplification. As the spectral topology is intimately connected with the non-Hermtian skin effect, self-acceleration offers a practical and sensitive dynamic signal for its detection, particularly in quantum systems where decoherence dominates at long times.

## Results

### Time-multiplexed quantum walk

We simulate the dynamics of a local excitation along a dissipative lattice using photonic quantum walks[57–59] (see "Methods" and the Supplementary Information Note 1). Taking the more general two-dimensional quantum walk as an example, we implement the non-unitary Floquet operator

$$U = M_y S_y R(\theta_2) M_x S_x R(\theta_1), \qquad (1)$$

where shift operators are defined as $S_j = \sum_{\boldsymbol{r}} |0\rangle\langle 0| \otimes |\boldsymbol{r} - \boldsymbol{e}_j\rangle\langle\boldsymbol{r}| + |1\rangle\langle 1| \otimes |\boldsymbol{r} + \boldsymbol{e}_j\rangle\langle\boldsymbol{r}|$ with $\boldsymbol{r} = (x, y) \in \mathbb{Z}^2$ labeling the coordinates of the lattice sites, $j \in \{x, y\}$, and $\boldsymbol{e}_x = (1, 0)$ and $\boldsymbol{e}_y = (0, 1)$. The shift operators move the walker in the corresponding directions, depending on the walker's internal degrees of freedom in the basis of $\{|0\rangle, |1\rangle\}$ (dubbed the coin states). The coin operator acts in the subspace of coin states $R(\theta_{1,2}) = \begin{pmatrix} \cos\theta_{1,2} & i\sin\theta_{1,2} \\ i\sin\theta_{1,2} & \cos\theta_{1,2} \end{pmatrix} \otimes \mathbb{1}_{\boldsymbol{r}}$, where $\mathbb{1}_{\boldsymbol{r}} = \sum_{\boldsymbol{r}} |\boldsymbol{r}\rangle\langle\boldsymbol{r}|$. The gain-loss operators are given by $M_j = \begin{pmatrix} e^{\gamma_j} & 0 \\ 0 & e^{-\gamma_j} \end{pmatrix} \otimes \mathbb{1}_{\boldsymbol{r}}$, which make the quantum walk non-unitary for finite $\gamma_x$ or $\gamma_y$. For the momentum-space Hamiltonian corresponding to $U$, see the Supplementary Information Note 1.

In the experiment, we encode the internal coin states $\{|0\rangle, |1\rangle\}$ in the photon polarizations $\{|H\rangle, |V\rangle\}$, and the external spatial modes through the discretized temporal shifts. For the latter, we build path-dependent time delays into the loop, so that the spatial superposition of the photonic walker is translated to the temporal superposition of multiple well-resolved pulses within each discrete time step[60]. To encode spatial modes in two dimensions, the temporal modes are further separated into two different time scales by the free-space

Mach−Zehnder interferometer: 80 ns in the $x$-dimension and 4.83 ns in the $y$-dimension. For detection, we record the arrival time of the photons using avalanche photodiodes with the help of an acoustic-optical modulator serving as an optical switch to remove undesired pulses[61].

In the quantum-walk dynamics, the time-evolved state at the end of each discrete time step $t$ is $|\psi(t)\rangle = U^t |\psi(0)\rangle = e^{-iHt}|\psi(0)\rangle$, where we define an effective Hamiltonian $H$. Apparently, the quantum walk implements a stroboscopic simulation of the Hamiltonian $H$ at integer time steps (see the Supplementary Information Note 1). We measure the center-of-mass position of the walker, defined through[56]

$$\boldsymbol{n}_{CM}(t) = \frac{\sum_{\boldsymbol{r}} \langle\psi(t)|\boldsymbol{r}|\psi(t)\rangle}{\sum_{\boldsymbol{r}} \langle\psi(t)|\psi(t)\rangle}. \qquad (2)$$

As illustrated in Fig. 1, starting from a local excitation, the motion of $\boldsymbol{n}_{CM} = (x_{CM}, y_{CM})$ is closely connected with the spectral geometry of the effective Hamiltonian $H$ on the complex plane. More explicitly, transforming $H$ to the momentum space, we have $H(\boldsymbol{k})|\psi_\pm(\boldsymbol{k})\rangle = E_\pm(\boldsymbol{k})|\psi_\pm(\boldsymbol{k})\rangle$, where $\boldsymbol{k}$ belongs to the first Brillouin zone, and $E_\pm(\boldsymbol{k})$ and $|\psi_\pm(\boldsymbol{k})\rangle$ are respectively the eigenenergies and eigenstates under the periodic boundary condition (PBC), with the subscripts $\pm$ indicating the band index. For a local initial state that is an equal-weight superposition of all eigenstates within a given band, for instance, $|\psi(0)\rangle = \sum_{\boldsymbol{k}} |\psi_+(\boldsymbol{k})\rangle \otimes |\boldsymbol{k}\rangle$, the short-time behavior of $\boldsymbol{n}_{CM}$ reads (see "Methods")

$$x_{CM}(t) \simeq \frac{1}{2} a_x t^2 - \frac{1}{2}, \quad y_{CM}(t) \simeq \frac{1}{2} a_y t^2 + \frac{1}{2}, \qquad (3)$$

where

$$a_x = \frac{1}{\pi^2} \int_{-\pi}^{\pi} \mathrm{d}k_x \mathrm{d}k_y E_I \frac{\partial E_R}{\partial k_x} = \frac{2}{\pi} \int_{-\pi}^{\pi} \mathrm{d}k_y \mathcal{A}_x(k_y),$$
$$a_y = \frac{1}{\pi^2} \int_{-\pi}^{\pi} \mathrm{d}k_x \mathrm{d}k_y E_I \frac{\partial E_R}{\partial k_y} = \frac{2}{\pi} \int_{-\pi}^{\pi} \mathrm{d}k_x \mathcal{A}_y(k_x). \qquad (4)$$

Here $E_R$ and $E_I$ are, respectively, the real and imaginary components of $E_+$. Importantly, $a_x$ and $a_y$ suggest that the short-time self-acceleration rate is proportional to the volume enclosed by the eigenspectrum of the corresponding band in the complex parameter space. An alternative understanding is that the self-acceleration rate is proportional to the averaged area enclosed by $E_+(k_x, k_y)$ on the complex plane as $k_y$ traverses the Brillouin zone, as shown in Fig. 1. It should be mentioned that, in non-Hermitian systems, self-acceleration of the wave function in the absence of external forces is a universal phenomenon observed for rather arbitrary excitations that are initially localized (see "Methods"). However, it is only when the system is initially prepared in an equal-weight superposition of all eigenstates within a given band, that the ensuing self-acceleration relates to the spectral geometry through $a_x$ and $a_y$.

### Simulating the one-dimensional dynamics

For our experimental demonstration, we first consider the case of one-dimensional quantum walks. In one-dimensional lattices, a general correspondence can be established between spectral geometry and self-acceleration. Such a correspondence is grounded on a general theorem given in the Supplementary Information Note 2, which extends previous theoretical results[56]. Basically, for suitable initial preparation of the system, the self-acceleration rate is proportional to the area enclosed by the PBC eigenspectrum of a given lattice band on the complex plane. Based on the general two-dimensional setup in Fig. 1, one-dimensional quantum walks can be realized by simply removing the free-space Mach−Zehnder interferometer within the

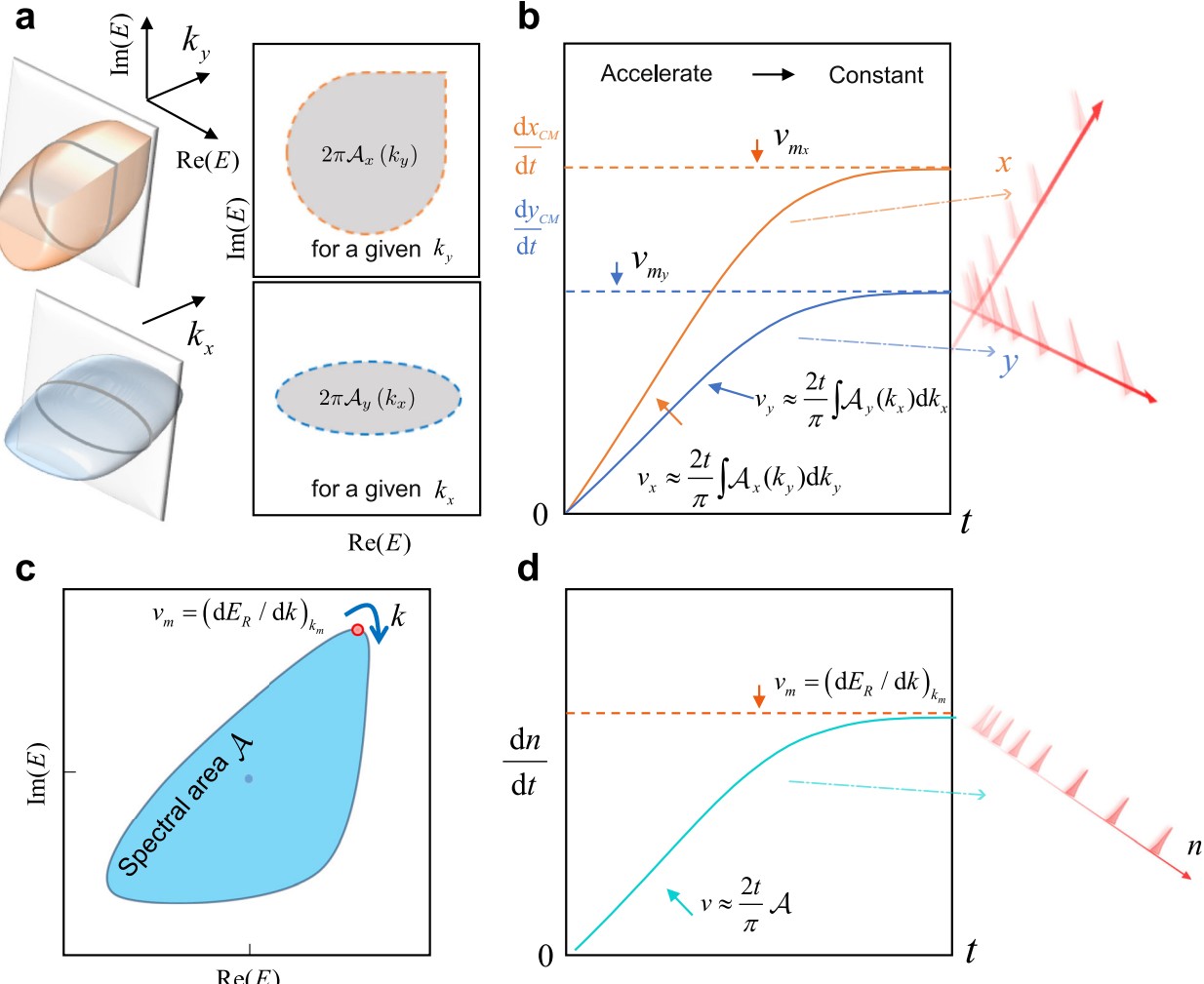

**Fig. 1 | Illustration of the connection between self-acceleration and spectral geometry. a** A schematic of the PBC energy spectra in a two-dimensional system with $k_y$ (upper panel) and $k_x$ (lower panel) as a parameter. **b** The corresponding two different volumes lead to distinct dynamic behaviors, i.e., different accelerated speeds of the motion of the wave packets in two directions, and eventually approach to a constant. **c** A schematic with a finite energy spectrum area in a one-dimensional system. **d** The motion of the wave packet shifts from accelerated to constant velocity as time increases.

loop. The resulting Floquet operator is given by

$$U = S_x M_x R(\theta). \qquad (5)$$

We focus on the parameter regime with $\theta$ being quite close to $\pi/2$ (we choose $\theta = 0.45\pi$ for experiment), where the complex eigenenergies of the effective Hamiltonian $H$ are approximately (see "Methods")

$$E_\pm(k) \approx \pm \frac{\pi}{2} \mp \cos\theta \cos(k - i\gamma_x), \qquad (6)$$

the last term of Eq. (6) corresponding to those of a typical Hatano–Nelson model, i.e., $H_{HN} = (1/2)\cos(\theta)e^{ik-\gamma_x} + (1/2)\cos(\theta)e^{-ik+\gamma_x}$. We initialize the system in the superposition state (representing a local excitation) $|\psi(0)\rangle_1 = e^{\gamma_x}|0\rangle \otimes |x = -1\rangle + |1\rangle \otimes |x = 0\rangle$, which is an equal-weight superposition of the Bloch states (in the first Brillouin zone) of $H$ with eigenenergy $E_+(k)$. The experimental implementation of such a local initial state, which is pivotal to our measurement scheme, is discussed in "Methods" and the Supplementary Information Note 3. The resulting short-time dynamics of $x_{CM}$ then follows that in Eq. (3), with the self-acceleration rate given by

$$a_x = \frac{2}{\pi} \int_{-\pi}^{\pi} dk_x E_I \frac{d}{dk} E_R := \frac{2}{\pi}\mathcal{A}. \qquad (7)$$

Here $\mathcal{A}$ corresponds to the area enclosed by the complex eigenenergy $E_+(k_x)$ in the complex plane, taken with the appropriate sign depending on the circulation direction of the PBC energy loop. Such a sign naturally indicates the direction of self-acceleration.

In Fig. 2a–c, we show the measured spatial population evolution of the dynamics under different gain-loss parameters $\gamma_x$. The wave-function propagation becomes asymmetric when the gain-loss parameter $\gamma_x$ becomes finite. In Fig. 2d–f, we show the measured $x_{CM}(t)$, which are quadratic in time when $\gamma_x \neq 0$, consistent with theoretical predictions. By fitting the center-of-mass propagation of the wave functions, we extract the quantity $\mathcal{A}$ from the self-acceleration rate (see Fig. 2g), which agrees well with the numerically calculated area enclosed by the eigenspectrum $E_+(k)$ on the complex plane (see Fig. 2h). The self-acceleration in the one-dimensional model is a clear signature of the non-Hermitian skin effect under the open-boundary condition. In fact, in systems that do not display the non-Hermitian skin effects, the PBC energy spectrum collapses to an open arc enclosing a vanishing area $\mathcal{A} = 0$, and thus acceleration vanishes according to Eq. (7).

### Simulating the two-dimensional dynamics

In two dimensions, we focus on the coin parameters close to $(\theta_1 = 0, \theta_2 = \pi/2)$, where the eigenenergies of the effective Hamiltonians

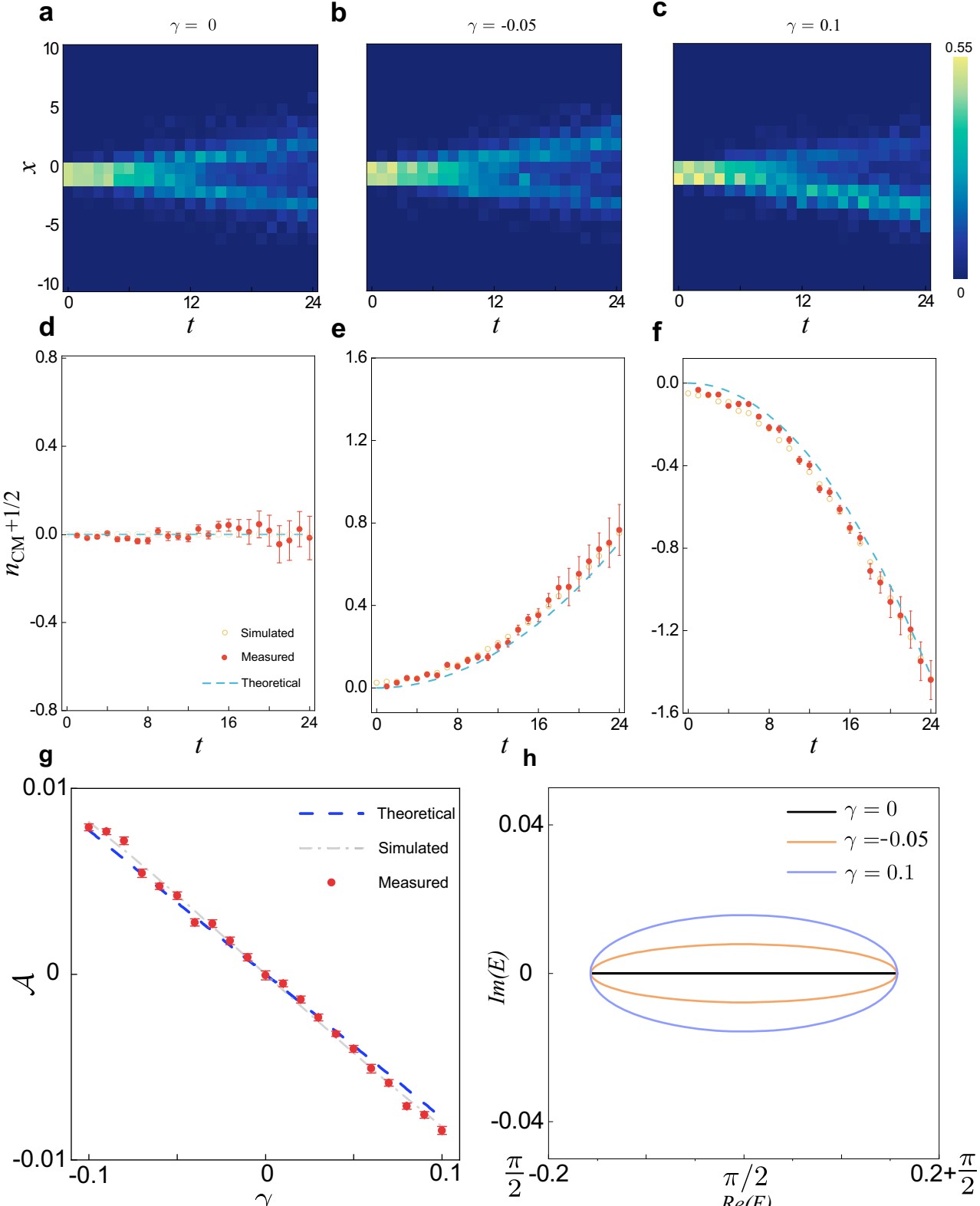

**Fig. 2 | Self-acceleration in one-dimensional dynamics. a–c** Dynamic evolutions governed by the effective non-Hermitian Hamiltonian with parameters $\gamma_x = 0$, $\gamma_x = -0.05$ and $\gamma_x = 0.1$, respectively. **d–f** Evolution of the center of mass $n_{CM}(t)$ as a function of the discrete time step $t$ corresponding to the dynamic evolutions in (**a–c**), respectively. **g** Areas enclosed by $E_+(k)$ versus the gain-loss parameter $\gamma = \gamma_x$. We take a local initial state $|\psi(0)\rangle_1$ and the coin parameter $\theta = 0.45\pi$. **h** PBC energy spectra for increasing values of $\gamma$.

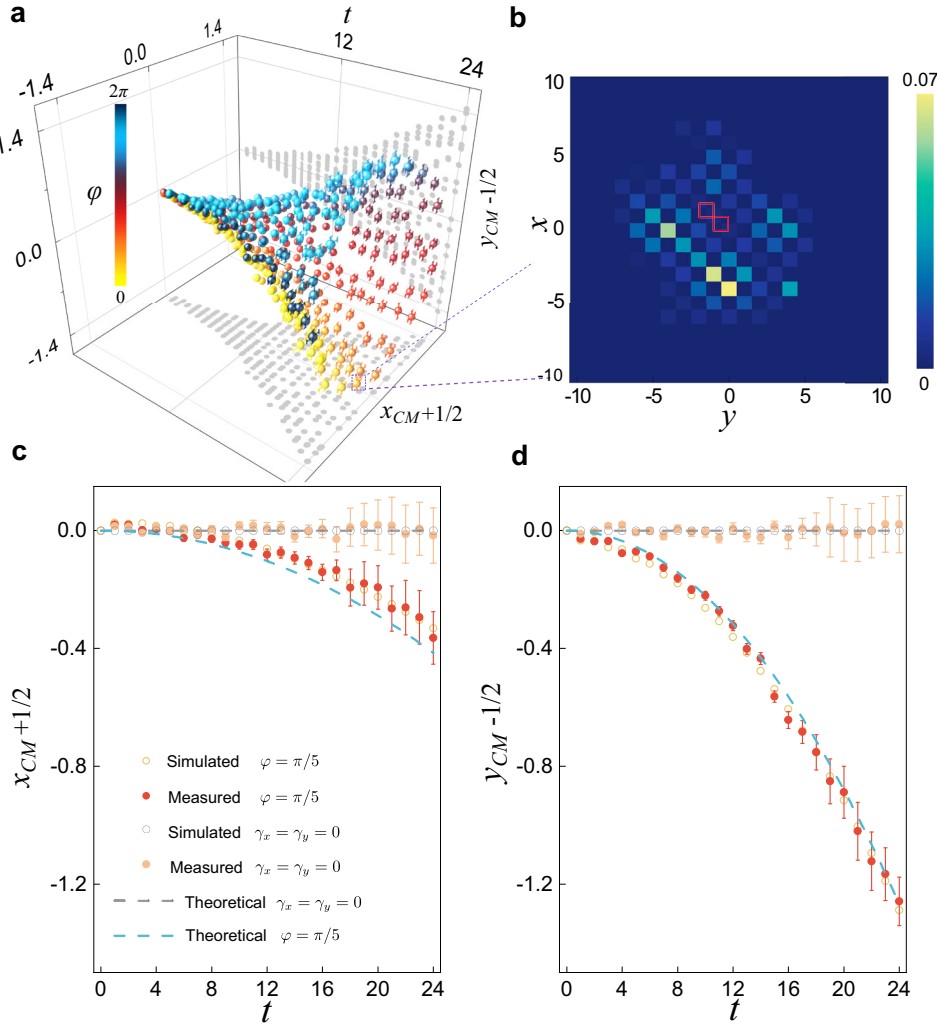

**Fig. 3 | Self-acceleration in two-dimensional dynamics. a** Wave packet center of mass $x_{CM}(t)$ ($y_{CM}(t)$) as a function of the discrete time step $t$. The gain-loss parameter is taken as $\gamma = 0.08$. **b** Probability distribution of the 24-time-step quantum walk with $\varphi = \pi/5$. We mark the position occupied by the initial state as the red square.

**c, d** The wave packet center of mass $x_{CM}(t)$ ($y_{CM}(t)$) as a function of the discrete time step $t$ with $\varphi = \pi/5$. The other parameters are $\theta_1 = 0.12$, $\theta_2 = \pi/2 - 0.12$, and the initial state $|\psi(0)\rangle_2$.

are

$$E_\pm(k_x, k_y) \approx \pm \frac{\pi}{2} \pm \tilde{E}(k_x, k_y). \tag{8}$$

Here $\tilde{E}(k_x, k_y) = \theta_1 \cos(k_y - i\gamma_y - k_x + i\gamma_x) - (\pi/2 - \theta_2) \cos(k_x - i\gamma_x + k_y - i\gamma_y)$ corresponds to single-band lattice on a two-dimensional square lattice, i.e., $\hat{H} = \sum_{n,m,k,l} H_{n,m,k,l} |n,m\rangle \langle k,l|$, with matrix Hamiltonian

$$
\begin{aligned}
H_{n,m,n-1,m+1} &= \theta_1 e^{(\gamma_y - \gamma_x)/2} \\
H_{n,m,n-1,m-1} &= (\theta_2 - \pi/2) e^{(-\gamma_y - \gamma_x)/2} \\
H_{n,m,n+1,m-1} &= \theta_1 e^{(\gamma_x - \gamma_y)/2} \\
H_{n,m,n+1,m+1} &= (\theta_2 - \pi/2) e^{(\gamma_y + \gamma_x)/2},
\end{aligned}
\tag{9}
$$

corresponding to the hopping amplitudes in four different directions. We initialize the system in the local state $|\psi(0)\rangle_2 = |0\rangle \otimes |x=0, y=0\rangle - e^{\gamma_x - \gamma_y} |1\rangle \otimes |x=-1, y=1\rangle$, which is a superposition of all the Bloch states corresponding to $E_+(k_x, k_y)$, again facilitated by the choice of the coin parameters.

In Fig. 3a, we show the time evolution of $(x_{CM}, y_{CM})$ under different gain-loss parameters $\gamma_x$ and $\gamma_y$, which are parameterized through

$\gamma_x = \gamma \cos \varphi$ and $\gamma_y = \gamma \sin \varphi$. Consistent with a previous study[59], the tuning of the parameters gives rise to directional propagation in the two-dimensional plane ($x$- and $y$-direction), which underlies the emergence of the non-Hermitian skin effect when open boundaries are enforced. An example of the full population evolution is illustrated in Fig. 3b. Apparently, for finite $\gamma_x$ or $\gamma_y$, the corresponding $x_{CM}$ or $y_{CM}$ exhibits quadratic behavior at early times, consistent with the predicted self-acceleration.

In Fig. 4, we explicitly demonstrate the correspondence between the spectral volume and the self-acceleration. For convenience, we focus on the case $\gamma_x = \gamma_y$, where the dynamics along the $x$ and $y$ directions are symmetric. Both the spectral volume and the fitted self-acceleration increase linearly with increasing $|\gamma_{x(y)}|$, consistent with the theoretical analysis. Similar to one-dimensional quantum walks, the self-acceleration is a precursor of persistent drift (or current) at long times and thus indicates the accumulation of excitation at the edges or corners of a finite two-dimensional domain (or equivalently, the appearance of the so-called non-reciprocal skin effect[16]). It should be mentioned that, in two-dimensional systems, the skin effect is a universal phenomenon that appears under rather arbitrary boundary shapes[16], and thus it persists even when the self-acceleration vanishes (see "Methods").

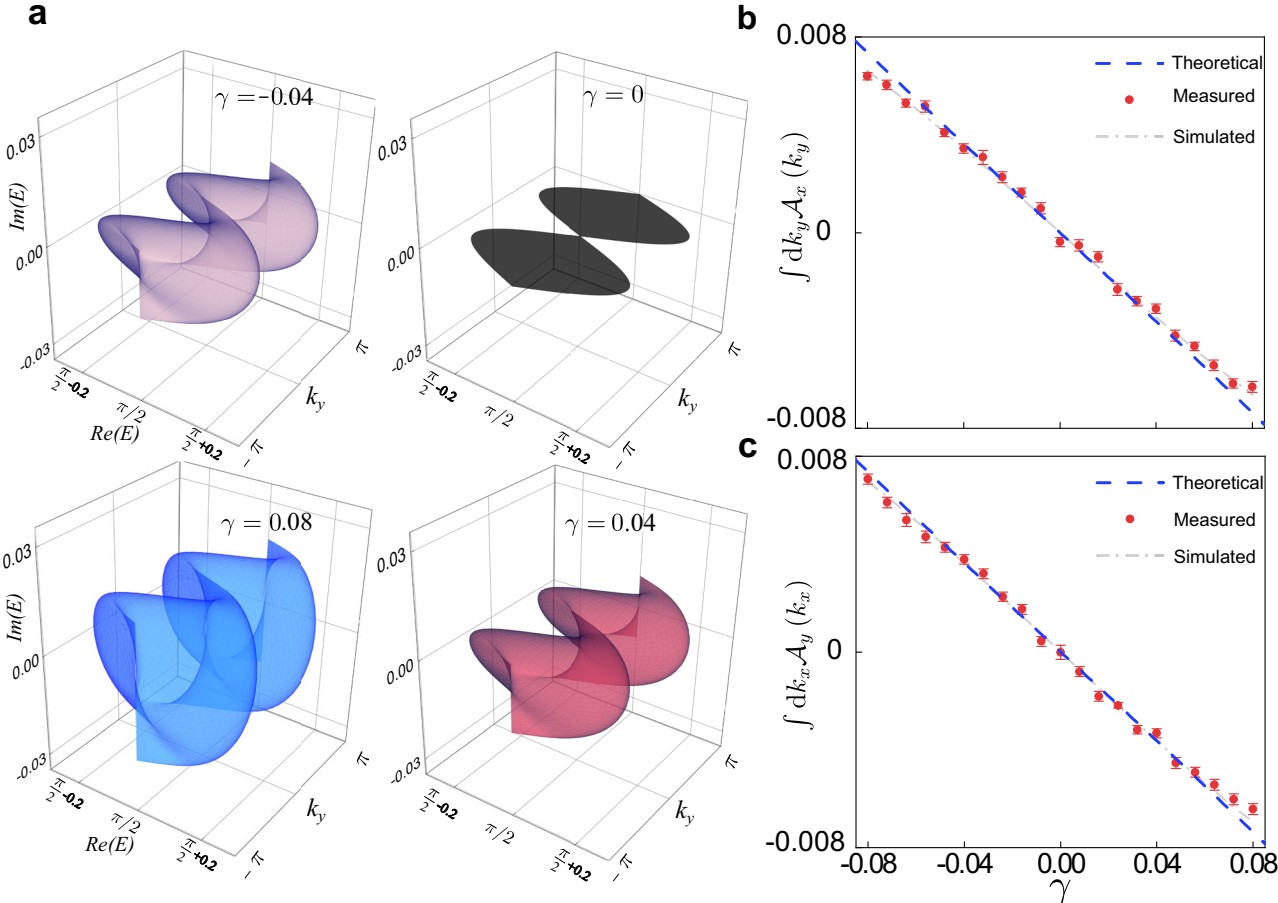

**Fig. 4 | Measurement of the spectral volume. a** PBC energy spectrum with $k_y$ as a parameter and for $\gamma_x = \gamma_y$. The black, purple, red, and blue surfaces correspond to $\gamma_{x(y)} = 0$, $\gamma_{x(y)} = -0.04$, $\gamma_{x(y)} = 0.04$, and $\gamma_{x(y)} = 0.08$, respectively. **b, c** The average of the areas enclosed by $E_+(k_x, k_y)$ as a function of $\gamma_{x(y)}$. The other parameters are $\theta_1 = 0.12$, $\theta_2 = \pi/2 - 0.12$, and the initial state $|\psi(0)\rangle_2$. The blue dashed line represents the theoretical results, and the gray dashed lines correspond the results fitted by experimental data.

## Crossover between short- and long-time dynamics

In the presence of a non-trivial spectral point-gap topology, it is well-established that the long-time dynamics of a local excitation is a directional propagation with a constant drift velocity[55,57,58], indicating that the self-acceleration ceases in the long-time evolution. The asymptotic drift velocities for two-dimensional dynamics are defined as the group velocity at the quasimomentum $(k_{m_x}, k_{m_y})$, with

$$v_{m_{x,y}} = \frac{dE_R(k_{m_x}, k_{m_y})}{d k_{x,y}} \quad (10)$$

where the eigenenergy $E_+(k_x = k_{mx}, k_y = k_{my})$ corresponds to the largest imaginary part of $E_+(k_x, k_y)$ (whose corresponding eigenmode survives at long times). As such, the combination of drift velocity at long times and self-acceleration at short times provides a complete correspondence between the spectral geometry and bulk dynamics of a local excitation.

In Fig. 5, we experimentally characterize the crossover from self-acceleration-dominated dynamics at short times (cyan dashed curves), to a flow at the drift velocity (purple dashed curves) at long times. This is achieved by choosing parameters such that the difference between self-acceleration and constant motion is appreciable at the experimentally accessible time steps.

## Discussion

Unveiling the correspondence between dynamical and spectral properties of classical and quantum systems is a fundamental problem and a major challenge in different areas of physics. While such a correspondence is quite well understood in closed systems described by Hermitian Hamiltonians, it remains largely unexplored for open systems. Using dissipative photonic quantum walks, here we have experimentally demonstrated a fundamental correspondence between the spectral geometry and the dynamics of local excitations in open systems described by effective non-Hermitian Hamiltonians, showing that a non-trivial spectral topology generally corresponds to a transient self-acceleration of the wave function. Our results provide major advancements in the understanding of the correspondence between spectral geometry and dynamics beyond the Hermitian paradigm and could stimulate further studies on an emergent area of research.

## Methods

### Self-acceleration for one-dimensional quantum walks

In the one-dimensional quantum walk, the Floquet operator reads $U = S_x M_x R(\theta)$, where $S_x = \sum_x |0\rangle\langle 0| \otimes |x-1\rangle\langle x| + |1\rangle\langle 1| \otimes |x+1\rangle\langle x|$ is the spatial shift operator, $M_x = \sum_x \begin{pmatrix} e^{\gamma_x} & 0 \\ 0 & e^{-\gamma_x} \end{pmatrix} \otimes |x\rangle\langle x|$ is the gain-loss operator with the gain-loss parameter $\gamma_x$, and $R(\theta) = \sum_x \begin{pmatrix} \cos\theta & i\sin\theta \\ i\sin\theta & \cos\theta \end{pmatrix} \otimes |x\rangle\langle x|$ is the coin operator. In the momentum space, the Floquet operator $U$ takes the form

$$U(k) = d_0\sigma_0 + id_x\sigma_x + id_y\sigma_y + id_z\sigma_z, \quad (11)$$

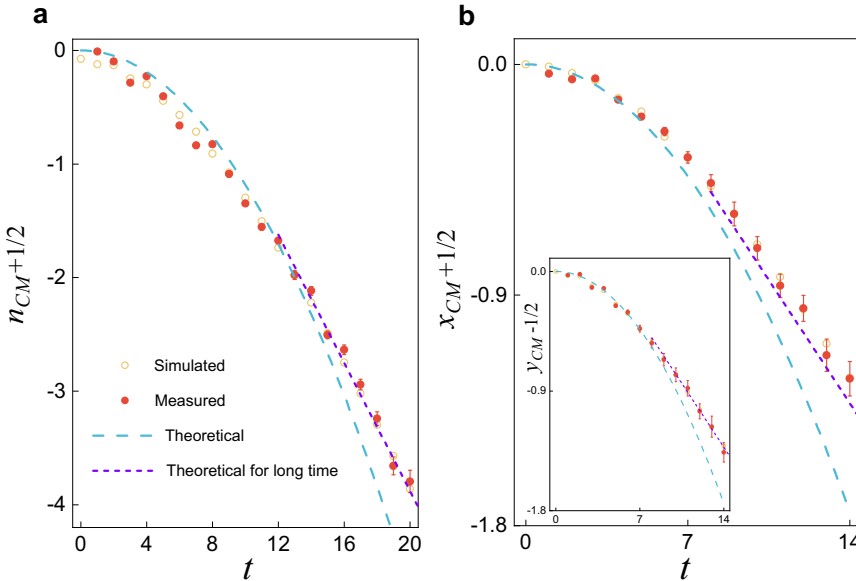

**Fig. 5 | Connecting the short- and long-time dynamics. a** Evolution of the center of mass $n_{CM}(t)$ as a function of the discrete-time step $t$ in the one-dimensional quantum walks with $\theta = 0.41\pi$ and $\gamma_x = 0.15$. We take a local initial state $|\psi(0)\rangle_1$.

**b** Wave packet center of mass $x_{CM}(t)$ ($y_{CM}(t)$) as a function of the discrete time step $t$ in the two-dimensional quantum walks with $\gamma_x = \gamma_y = 0.26$, $\theta_1 = 0.12$, $\theta_2 = \pi/2 - 0.12$ the initial state $|\psi(0)\rangle_2$.

where the expressions for $d_{x,y,z}$ are given in the Supplementary Information Note 1, and $\sigma_{x,y,z}$ are the Pauli matrices. Defining the momentum-space Hamiltonian $H_k$ through $U(k) = e^{-iH_k}$ (see the Supplementary Information Note 1), its quasienergies are given by

$$E_\pm(k) = \pm \arccos[\cos\theta \cos(k - i\gamma_x)], \tag{12}$$

with corresponding eigenstates (in the coin-state basis)

$$|\psi_\pm(k)\rangle = \begin{pmatrix} \frac{d_z \pm \sqrt{d_x^2 + d_y^2 + d_z^2}}{d_x + id_y} \\ 1 \end{pmatrix}. \tag{13}$$

In the experiment we set $\theta \approx \pi/2$, so that one has

$$\begin{aligned} E_\pm(k) &\simeq \pm\frac{\pi}{2} \mp \tilde{E}(k) \\ &= \pm\frac{\pi}{2} \mp \cos\theta \cos(k - i\gamma_x), \end{aligned} \tag{14}$$

and

$$|\psi_\pm(k)\rangle \simeq \begin{pmatrix} \pm e^{\gamma_x + ik} \\ 1 \end{pmatrix}. \tag{15}$$

Notice that $\tilde{E}(k)$ coincides with the dispersion of the Hatano–Nelson model with asymmetric nearest-neighbor hopping amplitudes $(1/2)\cos(\theta)e^{-\gamma_x}$ and $(1/2)\cos(\theta)e^{\gamma_x}$.

In our experiment, the quasi-local initial state can be expressed as

$$\begin{aligned} |\psi(0)\rangle_1 &= e^{\gamma_x}|0\rangle \otimes |x = -1\rangle + |1\rangle \otimes |x = 0\rangle \\ &= \frac{1}{2\pi}\int_{-\pi}^{\pi} dk |\psi_+(k)\rangle \otimes |k\rangle. \end{aligned} \tag{16}$$

Note that the sum over $k$ is approximated by integral at finite system size. The time-evolved state is then

$$|\psi(t)\rangle = \frac{(-i)^t}{2\pi}\int_{-\pi}^{\pi} dk \begin{pmatrix} e^{\gamma_x + ik} \\ 1 \end{pmatrix} e^{i\tilde{E}(k)t} \otimes |k\rangle, \tag{17}$$

and the center-of-mass of the normalized wave function is defined through[56]

$$n_{CM}(t) := \frac{\langle\psi(t)|x|\psi(t)\rangle}{\langle\psi(t)|\psi(t)\rangle}. \tag{18}$$

For short-time dynamics, making use of the truncated expansion $e^{2\tilde{E}_I(k)t} \approx 1 + 2\tilde{E}_I(k)t$, where $\tilde{E}(k) = \tilde{E}_R(k) + i\tilde{E}_I(k)$, after some straightforward calculations one obtains

$$n_{CM}(t) = \frac{\mathcal{A}}{\pi}t^2 - \frac{1}{2}. \tag{19}$$

Here $\mathcal{A} := \int_{-\pi}^{\pi} dk_x E_I \frac{d}{dk}E_R$ is the area enclosed by the complex quasienergy dispersion $E_+(k)$ in the complex plane.

In the long-time limit, the dynamics would be dominated by the Bloch mode $k_m$ where $E_I(k)$ is the global maximum. This corresponds to the conditions $\left(\frac{dE_I(k_m)}{dk}\right) = 0$ and $\left(\frac{d^2E_I(k_m)}{dk^2}\right) < 0$. Defining $\xi = k - k_m$, we expand the quasienergy around $k_m$, where the leading orders give $E(k) \simeq E(k_m) + \left(\frac{dE(k_m)}{dk}\right)\xi + \frac{1}{2}\left(\frac{d^2E(k_m)}{dk^2}\right)\xi^2$. It follows that

$$\begin{aligned} n_{CM}(t) &\approx \frac{2\pi t\left(\frac{dE_R(k_m)}{dk}\right)e^{2E_I(k_m)t}\int_{-\infty}^{\infty} d\xi\, e^{t\xi^2\left(\frac{d^2E_I(k_m)}{dk^2}\right)}}{2\pi e^{2E_I(k_m)t}\int_{-\infty}^{\infty} d\xi\, e^{t\xi^2\left(\frac{d^2E_I(k_m)}{dk^2}\right)}} \\ &= v_m t, \end{aligned} \tag{20}$$

where $v_m = \left(\frac{dE_R(k_m)}{dk}\right)$ is identified as the drift velocity.

## Self-acceleration for two-dimensional quantum walks

Following a similar procedure outlined in the previous section, we derive the center-of-mass motion of the wave functions for two-dimensional quantum walks.

We start from the Floquet operator $U$ in Eq. (1) and focus on the parameter regimes $\theta_1 \approx 0$ and $\theta_2 \approx \pi/2$. Under these conditions, the

momentum-space quasienergies are approximately $E_{\pm}\left(k_x, k_y\right) \simeq \pm\frac{\pi}{2} \pm \tilde{E}\left(k_x, k_y\right)$, where

$$
\begin{aligned}
\tilde{E}\left(k_x, k_y\right) = {} & \theta_1 \cos\left(k_y - i\gamma_y - k_x + i\gamma_x\right) \\
& - \left(\frac{\pi}{2} - \theta_2\right) \cos\left(k_x - i\gamma_x + k_y - i\gamma_y\right).
\end{aligned}
\tag{21}
$$

The corresponding eigenstates are

$$
|\psi_{\pm}(\boldsymbol{k})\rangle \simeq \begin{pmatrix} 1 \\ \mp e^{i(k_x - i\gamma_x - k_y + i\gamma_y)} \end{pmatrix}.
\tag{22}
$$

The local initial state is the uniform superposition of Bloch states in the upper band

$$
\begin{aligned}
|\psi(0)\rangle_2 &= |0\rangle \otimes |x=0, y=0\rangle - e^{\gamma_x - \gamma_y}|1\rangle \otimes |x=-1, y=1\rangle \\
&= \frac{1}{4\pi^2} \int_{-\pi}^{\pi} \int_{-\pi}^{\pi} dk_x\, dk_y |\psi_+(\boldsymbol{k})\rangle \otimes |\boldsymbol{k}\rangle.
\end{aligned}
\tag{23}
$$

We further write the time-evolved wave function at time $t$ as

$$
|\psi(t)\rangle = \sum_{x,y} \begin{pmatrix} (-i)^t \tilde{\psi}_{x,y}(t) \\ -(-i)^t \tilde{\psi}_{x+1, y-1}(t) \end{pmatrix} \otimes |x, y\rangle,
\tag{24}
$$

where

$$
\tilde{\psi}_{x,y}(t) = \frac{1}{(2\pi)^2} \int_{-\pi}^{\pi} \int_{-\pi}^{\pi} dk_x\, dk_y\, e^{ik_x x + ik_y y - i\tilde{E}(k_x, k_y)}.
\tag{25}
$$

Defining the center-of-mass positions of the wave function as in Eq. (2), we have $x_{CM}(t) \simeq \frac{1}{2} a_x t^2 - \frac{1}{2}$ and $y_{CM}(t) \simeq \frac{1}{2} a_y t^2 + \frac{1}{2}$ for short-time dynamics, where the self-accelerations are given by $a_x = \frac{1}{\pi^2} \int_{-\pi}^{\pi} dk_x dk_y E_I \frac{\partial E_R}{\partial k_x}$ and $a_y = \frac{1}{\pi^2} \int_{-\pi}^{\pi} dk_x dk_y E_I \frac{\partial E_R}{\partial k_y}$. Here we defined $E_+(k_x, k_y) = E_R + iE_I$.

Apparently, the self-accelerations have a simple geometric interpretation in terms of the energy spectrum $E_+(k_x, k_y)$ in the complex plane. For instance, the acceleration along the $x$ direction can be written as

$$
a_x = \frac{2}{\pi} \int_{-\pi}^{\pi} dk_y \mathcal{A}_x\left(k_y\right),
\tag{26}
$$

where

$$
\mathcal{A}_x\left(k_y\right) := \frac{1}{2\pi} \int_{-\pi}^{\pi} dk_x E_I \frac{\partial E_R}{\partial k_x}.
\tag{27}
$$

For a fixed value of $k_y$ (taken as a parameter), the expression $\int_{-\pi}^{\pi} dk_x E_I \frac{\partial E_R}{\partial k_x}$ is the area enclosed by the spectrum $E\left(k_x, k_y\right)$ in the complex plane as $k_x$ traverses Brillouin zone. Alternatively, taking the integration over $k_y$ into account, $a_x$ is proportional to the volume enclosed by $2\pi\mathcal{A}_x\left(k_y\right)$ in the parameter space, as $k_y$ traverses the Brillouin zone.

Finally, for the long-time dynamics, we have

$$
x_{CM}(t) \sim v_{m_x} t, \quad y_{CM}(t) \sim v_{m_y} t,
\tag{28}
$$

where $v_{m_x}$ and $v_{m_y}$ in Eq. (10) are the drift velocities, corresponding to the location of the global maximum of $E_I(k_x, k_y)$.

## Experimental scheme

We implement both one- and two-dimensional quantum walks by employing a time-multiplexed configuration, sending attenuated single-photon pulses (with a wavelength of 808 nm and a pulse width

of 88 ps) through a fiber network. While each full cycle around the fiber loop represents a discrete time step, the built-in optical elements within the loop, such as the half-wave plates, polarization beam splitters, quarter-wave plates, realize the time-evolution operator $U$ within each step. We encode external spatial modes through discretized temporal shifts, while internal coin-state degrees of freedom are encoded using photonic polarizations. With this experimental arrangement, we successfully carried out one-dimensional and two-dimensional quantum walks in the same experimental platform under various configurations (see the Supplementary Information Note 3).

In the scenario of initial state preparation, as detailed in the main manuscript, achieving the required initial excitation involves establishing an equally weighted superposition of Bloch eigenstates within a specific lattice band. However, our experimental constraints prevent the direct encoding of such a state, given its inclusion of excitations on both odd and even lattice sites beyond the capabilities of our setup. To address this limitation, it is crucial to recognize that during the evolution process, wave packets characterized by odd or even positions in the initial state do not interfere at each step. Leveraging this key property, we strategically divide our experiment into two distinct steps. In the first step, the initial excitation exclusively occupies either the even or odd lattice sites. Subsequently, we reconstruct the dynamics of the wave packet by capitalizing on the linearity inherent in the system (see the Supplementary Information Note 4).

## Origin of self-acceleration in non-Hermitian dynamics

In an Hermitian system, according to the Ehrenfest theorem, a wave packet cannot accelerate in the absence of any external force. However, this is not the case for non-Hermitian systems[13,22]. To illustrate this point, let us consider for example the single-particle dynamics on a one-dimensional lattice with Hamiltonian in the physical space $\hat{H} = \hat{T} + V(x)$, where $x$ is the lattice site position, $V(x)$ is the external potential, $\hat{T} = T(\hat{p}_x)$ is the kinetic energy operator, $\hat{p}_x = -i\partial_x$ is the momentum operator, and $T(p_x)$ is the energy dispersion curve of a given lattice band. For the standard Hatano–Nelson model, for instance, one has $T(p_x) = J\exp(ip_x + \gamma) + J\exp(-ip_x - \gamma)$, where $J\exp(\pm\gamma)$ are the asymmetric left/right hopping amplitudes.

For a given initial excitation of the system $|\psi_0\rangle$ at time $t = 0$, with $\langle\psi_0|\psi_0\rangle = 1$, the evolved wave function for early times is given by

$$
|\psi_t\rangle = \exp(-i\hat{H}t)|\psi_0\rangle \simeq \left(1 - it\hat{H} - \frac{t^2}{2}\hat{H}^2\right)|\psi_0\rangle.
\tag{29}
$$

From this equation, one can readily calculate the time evolution of the mean position $\langle x\rangle = \langle\psi_t|x|\psi_t\rangle / \langle\psi_t|\psi_t\rangle$, up to the order $\sim t^2$, and the corresponding initial acceleration, $a_x = (d^2\langle x\rangle / dt^2)_{t=0}$, which reads explicitly

$$
a_x = \langle 2\hat{H}^\dagger x\hat{H} - x\hat{H}^2 - \hat{H}^{\dagger 2}x\rangle_0 + 2\langle\hat{H}^\dagger - \hat{H}\rangle_0 \langle\hat{H}^\dagger x - x\hat{H}\rangle_0.
\tag{30}
$$

In the above equation, $\langle\hat{A}\rangle_0 \equiv \langle\psi_0|\hat{A}\psi_0\rangle$ denotes the mean value of any operator $\hat{A}$ over the initial state $|\psi_0\rangle$. Let us now assume that there is not any external force, $V(x) = 0$, so that the Hamiltonian contains the kinetic energy term solely, $\hat{H} = \hat{T}$. Using the generalized commutation relation $[x, F(\hat{p}_x)] = i(dF/dp_x)$ for any function $F(p_x)$ of the momentum operator, one obtains

$$
\begin{aligned}
a_x = {} & \langle x(2\hat{T}^\dagger\hat{T} - \hat{T}^2 - \hat{T}^{\dagger 2})\rangle_0 + 2i\langle\frac{\partial\hat{T}^\dagger}{\partial p_x}(\hat{T}^\dagger - \hat{T})\rangle_0 \\
& + 2\langle\hat{T}^\dagger - \hat{T}\rangle_0 \langle\hat{T}^\dagger x - x\hat{T}\rangle_0.
\end{aligned}
\tag{31}
$$

Clearly, in any Hermitian system $\hat{T}^\dagger = \hat{T}$, we necessarily have $a_x = 0$. Conversely, in a non-Hermitian system where $\hat{T}^\dagger \neq \hat{T}$, the acceleration $a_x$

is non-vanishing for rather arbitrary initial states $|\psi_0\rangle$, with its value dependent on the specific initial excitation.

## Self-acceleration and the non-Hermitian skin effect

In one-dimensional models, there is a one-to-one correspondence between the transient self-acceleration, under suitable initial excitation of the lattice, and the non-Hermitian skin effect. In fact, the skin effect, that is, the localization of a macroscopic number of eigenstates near the boundaries, appears rather generally whenever the Hamiltonian displays a point-gap topology in the PBC energy spectrum[14,15]. In the presence of the point-gap topology, the area $\mathcal{A}$, which is proportional to the self-acceleration, is necessarily non-vanishing. Additionally, we can have an intuitive explanation for self-acceleration from the skin effect. Let us consider a single-band system and single-site initial excitation of the lattice at $t = 0$. Clearly, the initial mean speed of the wave packet is zero, as all momenta $k$ within the Brillouin zone are equally excited. Owing to the non-Hermitian skin effect, Bloch modes displaying opposite group velocities are differently damped or amplified, and their unbalanced interference yields a deformation of the wave packet spreading and a directed transport at long times, characterized by a non-vanishing drift velocity $v_m$. This implies that the wave packet must necessarily undergo acceleration to attain a final non-zero speed. The resulting self-acceleration can be thus explained as an unbalanced interference effects between spectral wave packet components displaying opposite group velocities.

In two-dimensional systems, the non-Hermitian skin effect can depend on the geometry of the boundaries, and it is thus clear that the bulk dynamics of a wave packet alone (including transient self-acceleration and long-time drift motion), cannot uniquely determine the behavior of these boundary-dependent systems. Indeed, a recent work proved that, in higher dimensions, the non-Hermitian skin effect is a universal phenomenon observed for almost every local non-Hermitian Hamiltonian that displays a finite spectral area under the PBC, and when the shape of the open boundaries are taken without any special symmetries[16]. A distinction between generalized reciprocal and non-reciprocal skin effect has also been introduced[16], depending on whether the current in the system is vanishing or not, respectively.

In our two-dimensional non-Hermitian quantum walk, the non-vanishing self-acceleration clearly corresponds to a non-vanishing current, and thus the skin effect is of the latter type and is observable for arbitrary boundary shapes. The vanishing of the self-acceleration in a two-dimensional system does not necessarily imply the absence of the non-Hermitian skin effect under *arbitrary* shape of the boundaries, albeit it can disappear for a *specific* shape of the boundaries. To clarify this point, let us consider for example the two-dimensional square lattice described by the Bloch Hamiltonian

$$H(k_x, k_y) = J_x \cos k_x + i J_y \cos k_y, \qquad (32)$$

with real (Hermitian) hopping amplitude $J_x$ along the $x$ direction, and imaginary (non-Hermitian) hopping amplitude $iJ_y$ along the $y$ direction. Note that in this non-Hermitian model, the hopping amplitudes are reciprocal, and from the formulas of $a_x$ and $a_y$, it readily follows that $a_x = a_y = 0$, that is, transient self-acceleration in the bulk is absent. In this model, the non-Hermitian skin effect disappears for a square geometry of the boundaries due to the existence of two mirror symmetries. However, skin modes appear under different boundaries which break these mirror symmetries[16].

## Data availability

The data that support the findings of this study are available from the corresponding authors upon requests.

## Code availability

The codes that support the findings of this study are available from the corresponding authors upon requests.

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

## Acknowledgements

This work has been supported by the National Key R&D Program of China (Grant No. 2023YFA1406701) and National Natural Science Foundation of China (Grant Nos. 12025401, 92265209, 12374479, 11974331, 12104009 and 12104036).

## Author contributions

P.X. supervised the project, designed the experiments, analyzed the results, and wrote part of the paper. Q.L. performed the experiments with the contribution from K.K.W. and L.X. S.L. developed the theoretical aspects and revised the paper. W.Y. performed the theoretical analysis and wrote part of the paper.

## Competing interests

The authors declare no competing interests.
