## [Peer Review File · Nature Communications]

Self acceleration from spectral geometry in dissipative quantum-walk dynamicsREVIEWER COMMENTS

Reviewer #1 (Remarks to the Author):

The manuscript entitled „Self acceleration from spectral geometry in dissipative quantum-walk dynamics“ by Xue et al. describes photonic quantum walk experiments that display centre-of-mass acceleration for non-Hermitian wave-packet dynamics under periodic boundary conditions (PBC). The authors show that when the wave packet is initialized in an equal-weight superposition of eigenstates, the acceleration is given by the area enclosed by the complex PBC spectrum. From this, they claim that there is a clear correspondence between the effect of self acceleration and point gap topology - a connection that was first described in theory in Ref. [51] of the manuscript. Some of the results and short calculations in [51] are repeated in this manuscript to provide a self-contained presentation.

While I find the manuscript interesting and it touches on a very important question in the field (how non-Hermitian topology manifests itself in observable phenomena), I find there are a few unclear points in the manuscript which need to be clarified before I can make a recommendation about the manuscript's publication in Nat Commun as this may change the impact of the work. I also make a few suggestions to improve the presentation.

1) The authors frequently emphasize the generality of their results and talk about a „universal correspondence“, „general correspondence“ or „complete correspondence“ giving the impression that non-trivial spectral topology and self acceleration are in one-to-one correspondence with each other and can be used equivalently. In particular, they claim in the abstract that self acceleration is a „sensitive probe for phenomena in non-Hermitian systems that originate from spectral topology“.

However, there seem to be some important restrictions about when the connection between spectral topology and self acceleration holds. The authors themselves remark on one such restriction on page 4 where they state that in 2 dimensions the non-Hermitian skin effect can occur without observing self acceleration.

Furthermore, I suspect that self acceleration may in some cases occur without a non-trivial spectral topology which would imply that the observation of self acceleration is not sufficient to conclude that the spectral topology is non-trivial. I would ask the authors to consider the following counter example and clearly state in the manuscript when the

connection holds.

Counter example:

My suspicion is that self acceleration is in fact connected to the phenomenon of non-reciprocity. It is straightforward to suggest examples leading to non-reciprocal dynamics in which, however, there is no point gap and non-Hermitian skin effect under open boundary conditions (OBC). The authors claim on page 3 (without providing a reference or a proof) that in such cases, self acceleration vanishes, but I think it would be very important to check this for the following example.

For instance, a lattice model with symmetric left- and rightward nearest-neighbour hopping and next-nearest-neighbour hopping $J e^{i\phi}$ to the right but $J e^{-i\phi}$ to the left (and local decay at fixed rate for dynamical stability) will result in non-reciprocal transport under both PBC and OBC due to the interference between nearest- and next-nearest-neighbour couplings (the phase ϕ is gauge invariant and acts like an Aharonov-Bohm phase).

However, the spectrum does not have an open point gap (the eigenvalues fall on a line) and the system does not display the non-Hermitian skin effect. Nevertheless, the time evolution reveals directional propagation and the Green's function is asymmetric signalling non-reciprocity (indeed, the scattering response of such a system would be non-reciprocal). For further details on this example see Appendix B and Figure 10 of [SciPost Phys. 15, 173 (2023)].

The second example I would like to suggest is a model of a chain in which nearest neighbours are coupled via Hermitian hopping (rightward hopping $J e^{i\phi}$, leftward hopping $J e^{-i\phi}$) and additionally via an indirect coherent hopping at strengths J' via an auxiliary mode. This is the coherent model corresponding to the NH chain described in [Nat Commun 11, 3149 (2020)]. All modes experience decay and all modes except for the auxiliary modes experience local gain (modeled by a term $+i\kappa$ in the NH Hamiltonian). If the auxiliary modes are integrated out, the resulting NH chain is the Hatano-Nelson model with additional on-site decay which displays a point gap for asymmetric hopping (for ϕ different from $0, \pi$) and the non-Hermitian skin effect and displays the NH skin effect in which case one would, according to the manuscript, expect to see self acceleration.

However, if the auxiliary mode is not integrated out, there is no point gap and no non-Hermitian skin effect while the physical dynamics should be the same suggesting that self

acceleration without a point gap is possible.

To test the correspondence proposed by the authors, it would be very important to check (e.g. analytically or numerically) whether self acceleration occurs in the above case (which I suspect it might) and related cases. However, this would imply that the connection between spectral topology and self acceleration is not a one-to-one correspondence, so the observation of self acceleration would not be sufficient to deduce a non-trivial spectral topology and the claim of the abstract would not hold. In this case, the authors would need to clarify this point and choose a different term than „correspondence“ (e.g. „connection“) which otherwise implies an equivalence which may not necessarily be fulfilled in full generality. It would be ideal if the exact conditions for self acceleration could be given.

2) Some practical concern:

Ideally, one would like to be able to infer the spectral topology from a (more or less) simple experimental observation such as wave packet dynamics or scattering response. However, the phenomenon of self acceleration only seems to be straightforwardly connected to spectral topology when the wave packet is initialised in an equal-weight superposition of all eigenstates. Therefore, it seems that in order to test whether a system has a non-trivial spectral topology or not, one already has to know the Hamiltonian one would like to test (and then one could in fact simply calculate the spectrum leading to a somewhat circular argument). It would therefore be important to answer the question of what happens if we naively initialise in a different state that is uninformed by the Hamiltonian, e.g., a state that is localised either on one or the other sublattice on one site. Is it possible to make any general statements about self acceleration in such a general case? Is it still a correspondence or are there cases where self acceleration can occur without non-trivial topology and vice versa (see also point 2)?

I believe that it is very important to answer these questions to justify the claim of the abstract (that self acceleration is a „sensitive probe for phenomena in non-Hermitian systems that originate from spectral topology“).

3) Context:

In a series of works, a one-to-one correspondence was proven and demonstrated that allows to infer the spectral topology from a straightforward scattering experiment or the

steady state.

The phenomenon of directional amplification has been linked to spectral non-Hermitian topology [1-3] and has been successfully demonstrated in experiments [4]. This connection is very strong: indeed, there is a one-to-one correspondence between non-trivial non-Hermitian topology and directional amplification. Performing a simple scattering experiment is sufficient to determine whether the underlying system is non-trivial or not and extract the winding number by counting the number of amplifying channels [1-3]. This approach also does not require post-selection and the phenomenon is robust against disorder [2].

The authors may wish to mention at least [1] and [4] to put their work into a wider context.

[1] Nat Commun 11, 3149 (2020).

[2] Phys. Rev. Lett. 127, 213601 (2021).

[3] SciPost Phys. 15, 173 (2023).

[4] arXiv:2309.05825 (2023).

The effects described in this manuscript and in refs. [1-4] may to some extent be connected. In particular, some aspects mentioned by the authors (such as the conditions directional propagation and the fact that the sign of the winding number determines the direction of propagation) are also discussed in the works above.

Furthermore, the authors may find the intuitive explanation of directional amplification as a result of the accumulation of excitations at the system edge (see Figure 2 in [1]) helpful as they themselves make a related remark.

Note that directional amplification does not occur in the counter example mentioned under point 2 (only non-reciprocity), so it accurately predicts the topology also in that case [3].

4) I am a bit confused by role of dynamical stability. Clearly, the dynamics described here will not be dynamically stable (Figs. 1c and 2h) since some eigenstates correspond to eigenvalues with positive imaginary part. What are the consequences on the dynamics in the experiment when the spectrum encircles the origin and some states correspond to unstable modes? Does this imply the experiment can only be performed for a certain time duration (until the amplitudes are too large)? Do physical non-linear effects become relevant?

5) Presentation:

It would improve the accessibility of the manuscript for a wider audience if the authors would elaborate in a short paragraph on the connection between the implemented lattice model and the photonic quantum walk, i.e., the connection between Eq. (1) and a NH Hamiltonian. In particular, it would help the reader if the authors could explicitly state which lattice model is described by Eq. (1) and after equation (7) since it is not straightforward to infer from these equations and actually never explicitly stated. It would also be helpful to write H_k above Eq. (10). While there is more information about this in the Supplementary, it would improve the presentation if more details were given in the main text surrounding Eq. (1) and a reference to the Supplemental Material was given where due.

6) Exposition of the experiment:

Authors refer to Refs. [53-55] for the experimental techniques to implement a photonic quantum walk claiming that this is a well-established technique (although all cited papers and pre-prints date from 2022). However, given the broad nature of the journal it would help the readers if the authors could give a brief description of their setup (e.g. in the Methods section) and refer to it at appropriate places in the main text. The authors do describe their experimental setup in the Supplemental Material, however, this is currently left for the reader to discover so it would help the presentation if the Supplemental Material was referenced in the main text where appropriate.

7) When self acceleration is discussed it may be helpful to give an intuitive explanation for where the effect could come from. My intuition would be that due to the spectral topology some portions of the wave packet are damped more strongly than others resulting in the deformation of the wave packet which on the level of the centre-of-mass motion results in an acceleration. Perhaps the authors have another intuitive picture in mind. A possible place for such a discussion could be where self acceleration is first discussed in the Results section and/ or when the relation to the Ehrenfest theorem is discussed.

8) Connected to this: how does the shape of the wave packet change during the time evolution? Since the initial state is highly delocalised in k , I would expect strong wave-packet spreading. It would be my expectation that this spreading may after some time even make it

difficult to identify the centre of mass unambiguously (for some system parameters which result in fast spreading).

9) Some typos/ grammar improvements/ other comments:

Abstract:

- „Dynamic behaviours [...] originates“ -> „The dynamic behaviour [...] originate“
- „on the complex plane“ -> „in the complex plane“
- „the concomitant dynamics is“ -> „the concomitant dynamics are“
- „We the reveal similar correspondence“ -> „We the reveal a similar correspondence“

Introduction:

- „The dynamics of a system is intimately connected“ -> „The dynamics of a system are intimately connected“
- „Just as the energy of a celestial body impacts its trajectory [5], so the energy quantization accounts for the spontaneous collapse...“ -> „Just as the energy of a celestial body impacts its trajectory [5], the energy quantization accounts for the spontaneous collapse...“
- „one the complex plane“ -> „in the complex plane“
- „with non-trivial consequences in the system“ -> „with non-trivial consequences on the system“
- It would be helpful to cite a suitable subset of the following works together with [13-17], e.g. at the very least [R1] and [R4]:
[R1] Nat Commun 11, 3149 (2020).
[R2] Phys. Rev. Lett. 127, 213601 (2021).
[R3] SciPost Phys. 15, 173 (2023).
[R4] arXiv:2309.05825 (2023).

Results:

- Above Eq. (2): „Apparently, the quantum walk implements a stroboscopic simulation of the Hamiltonian H “ -> „The quantum walk implements a stroboscopic simulation of the Hamiltonian H [ref]“. Here, a citation and/ or reference to the Supplementary Material or Methods where this is shown would be helpful.
- Below Eq. (4): „with θ being quite close to $\pi/2$ “ -> it would be helpful to be more precise here.
- The authors state that the eigenvalues given in Eq. (5) resemble that of a Hatano-Nelson

model. However, the model they implement seems to have some sub-lattice structure so it would be helpful to precisely state which model is actually implemented (i.e., give either the lattice Hamiltonian or H_k).

- Below Eq. (5): „The experimental implementation of such a local initial state [...] is discussed in the Supplemental Material“ -> it would be helpful if more information about the experiment (perhaps just a brief exposition would be given in the main text or the Methods section.

- Below Eq. (7): specify the lattice model and/ or H_k that is implemented

- Below Eq. (7): „directional propagation in the two-dimensional plane“ -> Along x or y or both?

- Eq. (8): k_{mx} and k_{my} should be explicitly defined.

Methods:

- Eq. (9) and (10): it would be helpful to write H_k explicitly

- $|\Psi(k)\rangle$ in Eqs. (11), (13), and (15) are unnormalised. Since the normalisation constant in (15) will depend on k and the integral is taken over k , this will be important. The authors should check that their conclusions are correct once they include the k -dependent normalisation constant. The missing normalisation constant may also have an impact on Eq. (18).

- Eq. (14) and (15): for completeness, it should be mentioned that at finite system size the sum over k is approximated by integral.

- Above Eq. (19): it would be helpful to refer to the equation where $\tilde{E}(k_x, k_y)$ is defined.

- Page 6, right column: „Universality of self acceleration“ is a strong statement given the restrictions of the correspondence already remarked on by the authors.

- Above Eq. (27): „the evolved wave function in the early times“ -> „the evolved wave function for early times“

- Eq. (27): time dependent normalisation constant is missing.

Reviewer #2 (Remarks to the Author):

After carefully reading the work by P. Xue et al., entitled “Self acceleration from spectral geometry in dissipative quantum-walk dynamics,” I regret to inform that I cannot

recommend it for publication in Nature Communications.

My main concern is that the manuscript does not satisfy the publication criteria for Nature Communications. Specifically, the Aims & Scope state that “Papers published by the journal aim to represent important advances of significance to specialists within each field.”

Although the manuscript is well written, and the results are interesting; unfortunately, it does not provide significant advances, either theoretical or experimental, to the field which it belongs. To be more specific, I found a tremendous overlap between the theory presented in Ref. [51]: Phys. Rev. B 105, 245143 (2022), and the manuscript under review. Please note that the correspondence between “self-acceleration” and spectral topology is already discussed in the 2022 paper. On the other hand, the experimental setup used in the manuscript under review has been previously introduced in Nat. Commun. 14, 6283 (2023), meaning that new experimental tools/techniques are not demonstrated either.

From the discussion above, although I believe the experimental verification of the theoretical results predicted in Ref. [51] deserve to be published; it is my considered opinion that Nature Communications is not the proper journal.

Minor comment:

Although using the term “self-acceleration” might sound attractive, I believe the authors should explain the reader that this is not a true accelerating optical signal, but an interference pattern that results from the influence of the non-Hermiticity on the lattice modes. It is interesting to note from Figure 2 that the effect of the “gain-loss” parameter seems to be to unbalance the coin, thus making the quantum walk to move toward one specific direction. I wonder if changing the signs of γ in Fig. 2b and 2c makes the slopes of the plots change their direction as well.

Reviewer #3 (Remarks to the Author):

In the work titled "Self acceleration from spectral geometry in dissipative quantum-walk dynamics", the Authors demonstrate, theoretically and experimentally a short-time acceleration phenomenon associated with the spectral features of a non-hermitian process. In non-hermitian systems, the Hamiltonian has complex eigenvalues. This work shows that the dispersion relations of the energy's real and imaginary part are responsible for a parabolic trajectory of the center of mass (when considering the short time dynamics of initial states that fill uniformly a given band). This effect is clearly demonstrated in 1D and 2D quantum walks performed in the time bin degree of freedom. Both the theoretical and experimental results sound, novel (to the best of my knowledge) and can unveil new strategies to understand the physics of non-hermitian systems.

I thus recommend this work for publication in Nature Communciations.

Some minor comments,

- The authors may expand on the possible applications of their result. For example, they may discuss the possibility of monitoring self acceleration to probe some specific features of the system: is there any information about the topological phase that can be inferred? If initial states are prepared occupying more bands, are there possible interferometric protocol that can be devised, like in Landau-Zener-Stucklerberg interferometry?

Response to the review reports of the manuscript NCOMMS-23-46566 “Self acceleration from spectral geometry in dissipative quantum-walk dynamics”

Reply to Report of Referee #1

The manuscript entitled “Self acceleration from spectral geometry in dissipative quantum-walk dynamics” by Xue et al. describes photonic quantum walk experiments that display centre-of-mass acceleration for non-Hermitian wave-packet dynamics under periodic boundary conditions (PBC). The authors show that when the wave packet is initialized in an equal-weight superposition of eigenstates, the acceleration is given by the area enclosed by the complex PBC spectrum. From this, they claim that there is a clear correspondence between the effect of self acceleration and point gap topology - a connection that was first described in theory in Ref. [51] of the manuscript. Some of the results and short calculations in [51] are repeated in this manuscript to provide a self-contained presentation. While I find the manuscript interesting and it touches on a very important question in the field (how non-Hermitian topology manifests itself in observable phenomena), I find there are a few unclear points in the manuscript which need to be clarified before I can make a recommendation about the manuscripts publication in Nat Commun as this may change the impact of the work. I also make a few suggestions to improve the presentation.

We thank the Referee for finding the experiment interesting. In the following, we provide detailed responses to the Referee’s questions and suggestions. We trust that our comprehensive responses will meet the Referee’s expectations and satisfaction.

1A) The authors frequently emphasize the generality of their results and talk about a “universal correspondence”, “general correspondence” or “complete correspondence” giving the impression that non-trivial spectral topology and self acceleration are in one-to-one correspondence with each other and can be used equivalently. In particular, they claim in the abstract that self acceleration is a “sensitive probe for phenomena in non-Hermitian systems that originate from spectral topology”. However, there seem to be some important restrictions about

when the connection between spectral topology and self acceleration holds. The authors themselves remark on one such restriction on page 4 where they state that in 2 dimensions the non-Hermitian skin effect can occur without observing self acceleration. Furthermore, I suspect that self acceleration may in some cases occur without a non-trivial spectral topology which would imply that the observation of self acceleration is not sufficient to conclude that the spectral topology is non-trivial. I would ask the authors to consider the following counter example and clearly state in the manuscript when the connection holds.

We thank the Referee for raising this important question, which gives us the opportunity to better clarify some main points and extend our previous paper with additional theoretical analysis. The main question is basically: *under what conditions can a one-to-one correspondence between self-acceleration and spectral point-gap topology (or the NH skin effect) be stated ?*

We focus our reply to the 1D systems, since in 2D systems we already provided discussions in the main text. The equivalence between the spectral point-gap topology and the NH skin effect in 1D systems is established in [1], and thus is not discussed further in this reply.

1. Single-band lattice models

For the most exemplary class of NH lattice models, namely those featuring a single band with a point gap and no symmetry (the “counterexample” in [2] mentioned by the Referee belongs to this class), there is a one-to-one correspondence between the non-trivial spectral point gap topology and self-acceleration. Namely the following theorem 1 holds:

In any single-band NH lattice model, the early-time acceleration a_x of the center-of-mass wave packet under initial single-site excitation of the lattice is proportional to the spectral area \mathcal{A} enclosed by the complex PBC energy spectrum $H(k)$ in the complex plane, namely

$$a_x = \frac{2}{\pi} \mathcal{A}. \quad (\text{R1})$$

The proof of such a general theorem was given in the earlier theoretical paper [3] and it is therefore not given in the current experimental work. Note that the above result clearly provides a universal correspondence between self-acceleration and spectral topology in generic

one-band NH lattice systems without any special symmetry.

Some remarks are in order.

(i) The one-to-one correspondence holds provided that the system is initially excited in a single site. For different and arbitrary initial excitations of the system, as discussed in the Methods of our manuscript, self-acceleration is rather generally observed, even for systems with trivial point-gap spectral topology. These discussions give the complete condition for the emergence of the correspondence in 1D single-band models.

(ii) Since the initial excitation condition is independent of the Hamiltonian $H(k)$, the dynamical probing method provides a universal tool for testing the spectral topology of the system that does not require any *a priori* knowledge of the system Hamiltonian. Therefore our method does not have any practical limitations. Additionally, it only requires the system to maintain coherence for a short time, and hence it is favorable against decoherence or dephasing. Finally, we would like to mention that, as compared to other dynamical or spectral probing methods, such as those mentioned by the Referee in his/her point 3), our method not only provides a clear one-to-one correspondence between the non-trivial point gap topology and self-acceleration from the short-time wave packet evolution, but also yields a *geometric* information on the spectrum, namely its area \mathcal{A} .

(iii) Contrary to the Referee's argument in the suggested counter-example, our general theorem implies that, for a single-site initial excitation, self-acceleration is always vanishing in any one-band NH lattice system with trivial point-gap topology, even if the system is non-reciprocal (using the definition of non-reciprocity of Ref. [2]). See our reply to Referee 1's point 1B below.

2. Multi-band lattice models

The multi-band lattice models were not considered in the previous theoretical work [3]. In this case, a similar theorem exists which relates spectral geometry to self-acceleration, provided that some restrictions and appropriate initial conditions are satisfied. The following analysis is new and is now included in the revised Supplemental Material of our manuscript. Let us consider an M -band NH lattice system, so that each unit cell of the lattice comprises M sublattice sites. Let us indicate by $a_n(t)$, $b_n(t)$, $c_n(t)$, ... the amplitudes of the wave function at the M sites in the n -th unit cell of the lattice. Let $H(k)$ be the $M \times M$ Bloch

Hamiltonian of the system, and let us indicate by $E_\beta(k)$ and $(A_\beta(k), B_\beta(k), C_\beta(k), \dots)^T$, the eigenvalues and corresponding eigenvectors of $H(k)$ ($\beta = 1, 2, \dots, M$), respectively. Specifically, we have

$$E_\beta(k) \begin{pmatrix} A_\beta(k) \\ B_\beta(k) \\ C_\beta(k) \\ \dots \end{pmatrix} = H(k) \begin{pmatrix} A_\beta(k) \\ B_\beta(k) \\ C_\beta(k) \\ \dots \end{pmatrix}, \quad (\text{R2})$$

with $\beta = 1, 2, \dots, M$. Clearly, there is some arbitrariness in the choice of $A_\beta(k)$, $B_\beta(k)$, ..., since they are defined up to an arbitrary complex factor that may depend on the Bloch wave number. After a suitable choice of such a multiplication factor, it can be readily shown that the amplitudes $A_\beta(k)$, $B_\beta(k)$, ... can be chosen to satisfy the following conditions

$$|A_\beta(k)|^2 + |B_\beta(k)|^2 + |C_\beta(k)|^2 + \dots = 1, \quad A_\beta^*(k) \frac{dA_\beta}{dk} + B_\beta^*(k) \frac{dB_\beta}{dk} + \dots = 0. \quad (\text{R3})$$

for any $\beta = 1, 2, 3, \dots, M$. The first condition in Eq.(R3) corresponds to wave function normalization, whereas the second condition in Eq.(R3) corresponds to the gauge choice such that the diagonal elements of Berry connection vanish. Since the Hamiltonian $H(k)$ is non-Hermitian, it is rather general that the eigenenergies $E_\beta(k)$ and the corresponding eigenvectors are not single-valued functions of the Bloch wave number k , i.e., if we continuously follow the change of the eigenvalue $E_\beta(k)$, as k continuously changes from $k = -\pi$ to $k = \pi$ along the Brillouin zone, we can have a flip of eigenvalues and eigenvectors at the end of the cycle. Only after M cycles are the initial eigenenergy and eigenstate retrieved. The nontrivial energy-surface topology is typically associated with the fact that the cycle encloses one or more exceptional points (see for instance [4–7]). We say that the multi-band system has a trivial energy-surface topology whenever there is no eigenenergies and eigenvector flipping after one cycle (when k continuously change from $k = -\pi$ to $k = \pi$ along the Brillouin zone).

The mean position of a wave packet at time t in the physical space is given by

$$\langle n(t) \rangle = \frac{\sum_n n (|a_n(t)|^2 + |b_n(t)|^2 + |c_n(t)|^2 + \dots)}{\sum_n (|a_n(t)|^2 + |b_n(t)|^2 + |c_n(t)|^2 + \dots)}. \quad (\text{R4})$$

The following theorem 2 can be stated:

For a given band index α of the lattice ($\alpha = 1, 2, \dots, M$), let us prepare the system at the

initial time $t = 0$ in the state

$$a_n(0) = \frac{1}{2\pi} \int_{-\pi}^{\pi} A_\alpha(k) \exp(ikn) , b_n(0) = \frac{1}{2\pi} \int_{-\pi}^{\pi} B_\alpha(k) \exp(ikn) , c_n(0) = \frac{1}{2\pi} \int_{-\pi}^{\pi} C_\alpha(k) \exp(ikn) \dots \quad (\text{R5})$$

where the Bloch eigenvector amplitudes $A_\alpha(k)$, $B_\alpha(k)$, ... are assumed to satisfy Eq. (R3).

Then, if the system has a trivial energy-surface topology, the early-time acceleration a_x of the wave packet center of mass is given by

$$a_x = \frac{2}{\pi} \mathcal{A}_\alpha, \quad (\text{R6})$$

where \mathcal{A}_α is the area enclosed by the curve $E_\alpha(k)$ in the complex plane.

Proof. To prove theorem 2, let us first observe that the most general solution to the time-dependent Schrödinger equation in the multi-band lattice system reads

$$\begin{pmatrix} a_n(t) \\ b_n(t) \\ c_n(t) \\ \dots \end{pmatrix} = \sum_{\beta=1}^M \int_{-\pi}^{\pi} dk F_\beta(k) \begin{pmatrix} A_\beta(k) \\ B_\beta(k) \\ C_\beta(k) \\ \dots \end{pmatrix} \exp[ikn - iE_\beta(k)], \quad (\text{R7})$$

where the spectral amplitudes $F_\beta(k)$ are determined by the initial excitation of the system at time $t = 0$.

Let us assume that the initial state is the one defined by Eq.(R5), which corresponds to the choice $F_\beta(k) = (1/2\pi)\delta_{\alpha,\beta}$ for the spectral amplitude. Hence one has

$$\begin{pmatrix} a_n(t) \\ b_n(t) \\ c_n(t) \\ \dots \end{pmatrix} = \frac{1}{2\pi} \int_{-\pi}^{\pi} dk \begin{pmatrix} A_\alpha(k) \\ B_\alpha(k) \\ C_\alpha(k) \\ \dots \end{pmatrix} \exp[ikn - iE_\alpha(k)]. \quad (\text{R8})$$

From Eq.(R8), one can calculate the terms $\sum_n |a_n(t)|^2$, $\sum_n |b_n(t)|^2$, $\sum_n |c_n(t)|^2, \dots$ and $\sum_n n |a_n(t)|^2$, $\sum_n n |b_n(t)|^2$, $\sum_n n |c_n(t)|^2$. For example, one has

$$\sum_n |a_n(t)|^2 = \frac{1}{4\pi^2} \int_{-\pi}^{\pi} dk \int_{-\pi}^{\pi} dk' A_\alpha(k) A_\alpha^*(k') S(k - k') \exp[-itE_\alpha(k)t + iE_\alpha^*(k')t], \quad (\text{R9})$$

where we set

$$S(k - k') = \sum_n \exp[i(k - k')n]. \quad (\text{R10})$$

It follows that, in the range of variability of k and k' ,

$$S(k - k') = 2\pi\delta(k - k'), \quad (\text{R11})$$

and

$$\sum_n |a_n(t)|^2 = \frac{1}{2\pi} \int_{-\pi}^{\pi} dk |A_\alpha(k)|^2 \exp[2E_{I\alpha}(k)t], \quad (\text{R12})$$

where $E_{I\alpha}(k)$ is the imaginary part of the complex energy $E_\alpha(k)$. Similar expressions are found for $\sum_n |b_n(t)|^2$, $\sum_n |c_n(t)|^2$, ..., with

$$\sum_n |b_n(t)|^2 = \frac{1}{2\pi} \int_{-\pi}^{\pi} dk |B_\alpha(k)|^2 \exp[2E_{I\alpha}(k)t], \quad \sum_n |c_n(t)|^2 = \frac{1}{2\pi} \int_{-\pi}^{\pi} dk |C_\alpha(k)|^2 \exp[2E_{I\alpha}(k)t], \quad \dots \quad (\text{R13})$$

Taking Eq. (R3) into account, from Eqs. (R12) and (R13), one then obtains

$$\sum_n (|a_n(t)|^2 + |b_n(t)|^2 + |c_n(t)|^2 + \dots) = \frac{1}{2\pi} \int_{-\pi}^{\pi} dk \exp[2E_{I\alpha}(k)t]. \quad (\text{R14})$$

Let us now calculate $\sum_n n|a_n(t)|^2$, with

$$\sum_n n|a_n(t)|^2 = -i \frac{1}{4\pi^2} \int_{-\pi}^{\pi} dk' \int_{-\pi}^{\pi} dk A_\alpha(k) A_\alpha^*(k') \exp[-itE_\alpha(k)t + iE_\alpha^*(k')t] \cdot \frac{\partial S(k - k')}{\partial k} \quad (\text{R15})$$

After integration by parts and taking into account that $E_\alpha(\pi) = E_\alpha(-\pi)$ and $A_\alpha(\pi) = A_\alpha(-\pi)$ owing to the trivial energy surface topology, one readily obtains

$$\sum_n n|a_n(t)|^2 = \frac{i}{2\pi} \int_{-\pi}^{\pi} dk \left(A_\alpha^* \frac{dA_\alpha}{dk} - it|A_\alpha(k)|^2 \frac{dE_\alpha}{dk} \right) \exp[2E_{I\alpha}(k)t]. \quad (\text{R16})$$

Similar expressions are found for $\sum_n n|b_n(t)|^2$, $\sum_n n|c_n(t)|^2$, ..., with

$$\begin{aligned} \sum_n n|b_n(t)|^2 &= \frac{i}{2\pi} \int_{-\pi}^{\pi} dk \left(B_\alpha^* \frac{dB_\alpha}{dk} - it|B_\alpha(k)|^2 \frac{dE_\alpha}{dk} \right) \exp[2E_{I\alpha}(k)t], \\ \sum_n n|c_n(t)|^2 &= \frac{i}{2\pi} \int_{-\pi}^{\pi} dk \left(C_\alpha^* \frac{dC_\alpha}{dk} - it|C_\alpha(k)|^2 \frac{dE_\alpha}{dk} \right) \exp[2E_{I\alpha}(k)t], \\ &\dots \dots \end{aligned} \quad (\text{R17})$$

Taking into account of Eq. (R3), from Eqs.(R16) and (R17) one obtains

$$\sum_n n(|a_n(t)|^2 + |b_n(t)|^2 + |c_n(t)|^2 + \dots) = \frac{t}{2\pi} \int_{-\pi}^{\pi} dk \frac{dE_\alpha}{dk} \exp[2E_{I\alpha}(k)t]. \quad (\text{R18})$$

Using Eqs. (R14) and (R18), the evolution of the wave packet center of mass [Eq.(R4)] finally reads

$$\langle n(t) \rangle = \frac{t \int_{-\pi}^{\pi} dk \frac{dE_\alpha}{dk} \exp[2E_{I\alpha}(k)t]}{\int_{-\pi}^{\pi} dk \exp[2E_{I\alpha}(k)t]}, \quad (\text{R19})$$

which holds at any given instant t . In particular, in the early time dynamics with $t \rightarrow 0$, the above expression can be written as a series expansion in powers of t . At the lowest order, one obtains

$$\langle n(t) \rangle \simeq \frac{t^2}{\pi} \mathcal{A}_\alpha, \quad (\text{R20})$$

where we have set

$$\mathcal{A}_\alpha = \int_{-\pi}^{\pi} dk \frac{dE_{R\alpha}}{dk} E_{I\alpha}(k), \quad (\text{R21})$$

and $E_{R\alpha}(k)$, $E_{I\alpha}(k)$ are the real and imaginary parts of the energy $E_\alpha(k)$, respectively. Clearly, \mathcal{A}_α corresponds to the area enclosed by the energy $E_\alpha(k)$ in the complex plane as k traverses the Brillouin zone. Equation (R20) shows that, at early time, the wave packet center of mass displays an accelerated motion with an acceleration $a_x = 2\mathcal{A}_\alpha/\pi$, which thus provides a dynamical measure of the spectral geometry of the α -th lattice band. This concludes the proof of theorem 2.

Remark. Clearly, as compared to the single-band models, in the multi-band case the initial state preparation of the system can be more challenging in practice. Most importantly, as correctly pointed out by the Referee, it requires some knowledge of the Hamiltonian of the system. Nevertheless, for narrow-band systems with wide gaps (which corresponds for example to the case where the detuning of the on-site potentials at sites a , b , c , ... are much larger than the hopping amplitudes), the required initial excitation basically reduces to single-site excitation as in the single-band case, thus relaxing the condition of prior knowledge.

In our experiment, we are dealing with a two-band system, however as discussed in the Methods section of our manuscript, for a coin angle θ close to $\pi/2$ (which is the narrow-band regime), the corresponding initial excitation is very simple and we can accurately estimate the spectral areas from wave-packet self-acceleration measurements.

1B) My suspicion is that self acceleration is in fact connected to the phenomenon of non-reciprocity. It is straightforward to suggest examples leading to non-reciprocal dynamics in which, however, there is no point gap and non-Hermitian skin effect under open boundary conditions (OBC). The authors claim on page 3 (without providing a reference or a proof) that in such cases, self acceleration vanishes, but I think it would be very important to check this for the following example. For instance, a lattice model with symmetric left- and rightward

nearest-neighbour hopping and next-nearest-neighbour hopping $J \exp(i\phi)$ to the right but $J \exp(-i\phi)$ to the left (and local decay at fixed rate for dynamical stability) will result in non-reciprocal transport under both PBC and OBC due to the interference between nearest- and next-nearest-neighbour couplings (the phase ϕ is gauge invariant and acts like an Aharonov-Bohm phase). However, the spectrum does not have an open point gap (the eigenvalues fall on a line) and the system does not display the non-Hermitian skin effect. Nevertheless, the time evolution reveals directional propagation and the Green's function is asymmetric signalling non-reciprocity (indeed, the scattering response of such a system would be non-reciprocal). For further details on this example see Appendix B and Figure 10 of [SciPost Phys. 15, 173 (2023)].

FIG. R1. (a) Energy spectrum under PBC of the Hamiltonian Eq. (R22). The parameters are $\gamma = 0.3$, $\kappa = 1$, $J = 0.5$, and $\phi = \pi/2$. (b) Wave packet dynamics (snapshot on a pseudocolor map of site occupation probabilities $|a_n(t)|^2 / \sum_n |a_n(t)|^2$ versus time t) for single -site excitation of the lattice at site $n = 0$. (c) Corresponding temporal evolution of the wave packet center of mass $\langle n(t) \rangle$.

We have considered the counterexample suggested by the Referee. The model is a single-band lattice described by the Bloch Hamiltonian

$$H(k) = -i\gamma + 2\kappa \cos k + 2J \cos(2k + \varphi), \quad (\text{R22})$$

where γ is the on-site loss rate, κ is the hopping amplitude between the nearest-neighbor sites, $J \exp(\pm\varphi)$ are the hopping amplitudes between the next-nearest-neighbor sites, and φ is the gauge-invariant (Aharonov-Bohm or Peierls) phase. Since $H(-k) \neq H(k)$ for any hopping phase $\varphi \neq 0, \pi \pmod{2\pi}$, the system is not reciprocal but normal (according to the

definitions of Ref.[2]). The energy spectrum is clearly a straight horizontal segment in the complex energy plane, and it is thus topologically trivial [c.f. Fig.R1(a) and Fig.10(a) of Ref.[2]]. According to the general theorem 1 for single-band NH models stated above (our Reply to Referee 1 point A1), the self-acceleration of the wave packet vanishes for an initial single-site excitation, even though the system is not reciprocal. This can be readily checked by numerical simulations of the wave-packet propagation [see Fig.R1(b,c)]. Apparently, even though the wave packet spreading in the lattice is asymmetric around the initially-excited site $n = 0$ [Fig.R1(b)], the wave packet center of mass remains locked at $n = 0$ [see Fig.R1(c)]. Therefore, self-acceleration is not related to non-reciprocity, but rather to a non-trivial spectral point-gap topology.

1C) The second example I would like to suggest is a model of a chain in which nearest neighbours are coupled via Hermitian hopping (rightward hopping $J \exp(i\phi)$, leftward hopping $J \exp(-i\phi)$ and additionally via an indirect coherent hopping at strengths J' via an auxiliary mode. This is the coherent model corresponding to the NH chain described in [Nat Commun 11, 3149 (2020)]. All modes experience decay and all modes, except for the auxiliary modes, experience local gain (modeled by a term $+i\kappa$ in the NH Hamiltonian). If the auxiliary modes are integrated out, the resulting NH chain is the Hatano-Nelson model with additional on-site decay which displays a point gap for asymmetric hopping (for ϕ different from $0, \pi$) and the non-Hermitian skin effect and displays the NH skin effect in which case one would, according to the manuscript, expect to see self acceleration. However, if the auxiliary mode is not integrated out, there is no point gap and no non-Hermitian skin effect while the physical dynamics should be the same suggesting that self acceleration without a point gap is possible.

We have considered the second counter-example suggested by the Referee. As we understand from the textual report of the Referee and after looking at Ref. [8], the Referee asks us to basically consider the coherent model of the NH chain described in Ref. [8], but without driving and with auxiliary modes (responsible for indirect hopping) not eliminated from the analysis. The model, in the framework of an effective NH Hamiltonian described in the textual report of the Referee, is illustrated in Fig.R2(a). In the physical space, the

time-dependent Schrödinger equation of the model reads

$$i \frac{da_n}{dt} = -i\gamma a_n + J \exp(i\phi) a_{n+1} + J \exp(-i\phi) a_{n-1} + J'(b_n + b_{n-1}), \quad (\text{R23})$$

$$i \frac{db_n}{dt} = J'(a_n + a_{n+1}) - i\gamma' b_n, \quad (\text{R24})$$

where $J \exp(\pm i\phi)$ are the left/right hopping amplitudes between adjacent sites in the main sublattice A with effective on-site loss rate γ , J' is the hopping rate with the auxiliary sites of sublattice B with the loss rate γ' , and ϕ is the Aharonov-Bohm phase. The difference $\kappa = \gamma' - \gamma > 0$ provides the effective gain in the main modes of sublattice A. This is a two-band system with the Bloch Hamiltonian

$$H(k) = \begin{pmatrix} -i\gamma + 2J \cos(k + \phi) & J' + J' \exp(-ik) \\ J' + J' \exp(ik) & -i\gamma' \end{pmatrix}. \quad (\text{R25})$$

We thus apply our theorem 2 stated above (our Reply to Referee 1 point A1), and $\gamma = 0$, with the effective gain κ balancing the loss γ' .

As correctly stated by the Referee, in the large dissipative limit $\gamma' \gg J, J', \gamma$, one can

FIG. R2. (a) Schematics of the two-band lattice model described by Eqs. (R23) and (R24). (b-g) Energy spectra under the PBC of the Bloch Hamiltonian [Eq. (R25)], for parameters $\gamma = 0$, $J = 0.3$, $J' = 1$, $\phi = \pi/4$, and a few decreasing values of loss rate γ' in the auxiliary modes: $\gamma' = 8, 6, 4, 1, 0.5, 0.2$ from (b) to (g).

adiabatically eliminate the auxiliary-mode amplitudes b_n from Eqs. (R23) and (R24), by

letting $b_n(t) \simeq -i(J'/\gamma')(a_n + a_{n+1})$, which yields an effective single-band lattice described by the Schrödinger equation

$$i \frac{da_n}{dt} = -i \left(\gamma + \frac{2J'^2}{\gamma'} \right) a_n + \left(J \exp(i\phi) - i \frac{J'^2}{\gamma'} \right) a_{n+1} + \left(J \exp(-i\phi) - i \frac{J'^2}{\gamma'} \right) a_{n-1}. \quad (\text{R26})$$

For $\phi \neq 0, \pi$, similar to the Hatano-Nelson model, the left/right hopping amplitudes have different strengths, signaling the non-reciprocal transport and the appearance of the NH skin effect under the PBC. Reduced equations (R26) are in fact analogous to the mean-field equations (3) of Ref. [8], which display the NH skin effect. In this limit, the two energy bands of the Bloch Hamiltonian $H(k)$ under the PBC describe two well-separated closed loops: a large loop directly corresponding to Eq. (R26), and a narrow loop centered around the energy $E = -i\gamma'$ of strongly-damped modes; see Fig. R2(b,c). The model Eq. (R26) shows a nontrivial point-gap topology, corresponding to the NH skin effect under the OBC, and self-acceleration in the bulk under single-site excitation of the lattice. The acceleration rate is related to the spectral area of the large loop according to Eq. (R1). As the dissipation γ' is decreased and the adiabatic elimination starts to fail, we are dealing with a truly non-separable two-band model. In this regime a clear change in the spectral shape of energy curves is observed as γ' is decreased [see Fig.R2(b-g)]. We remark that we exclude from the analysis parameters where a nontrivial energy-surface topology arises, since in this case we cannot apply safely theorem 2. As Fig. R2 clearly shows, we still have a point gap topology even when we do not eliminate the auxiliary modes from the analysis, and, according to [1], we still have the NH skin effect. Therefore, the Referee's statement that the NH skin should disappear, when the auxiliary modes are integrated out, seems from our analysis an incorrect statement. For this model, we have directly checked the persistence of the NH skin effect in all cases (even when the energy-surface topology is nontrivial) by numerical computation of the Hamiltonian eigenstates in a lattice with open boundaries comprising $N = 100$ unit cells; our results clearly show a preferential localization to one edge of the lattice.

1D) To test the correspondence proposed by the authors, it would be very important to check (e.g. analytically or numerically) whether self acceleration occurs in the above case (which I suspect it might) and related cases. However, this would imply that the connection between spectral topology and self acceleration is not a one-to-one correspondence, so the observation of self acceleration would

not be sufficient to deduce a non-trivial spectral topology and the claim of the abstract would not hold. In this case, the authors would need to clarify this point and choose a different term than "correspondence" (e.g. "connection") which otherwise implies an equivalence which may not necessarily be fulfilled in full generality. It would be ideal if the exact conditions for self acceleration could be given.

Again, we are grateful to the Referee for raising such an important point, which gives us the opportunity to improve and expand the theoretical part of our work. We believe that we have properly addressed and fully clarified the Referee's concerns, namely, the validity of our claim about the correspondence between non-trivial point-gap spectral topology and self-acceleration in one-dimensional lattices. The correspondence is basically established via the rigorous theorems 1 and 2 enunciated above for single-band and multi-band lattice models. In two dimensions, the correspondence is conditional, as we already discussed in the previous manuscript. In the Supplementary Material of the revised manuscript we have added an entirely new section presenting the rigorous results, which are beyond the previous theoretical work [3] by one of the present authors.

Further, following the Referee's suggestion, we explicitly point out the correspondence is rigorous in one dimension, but also applicable to a wide class of two-dimensional models.

2) Some practical concern: Ideally, one would like to be able to infer the spectral topology from a (more or less) simple experimental observation such as wave packet dynamics or scattering response. However, the phenomenon of self acceleration only seems to be straightforwardly connected to spectral topology when the wave packet is initialised in an equal-weight superposition of all eigenstates. Therefore, it seems that in order to test whether a system has a non-trivial spectral topology or not, one already has to know the Hamiltonian one would like to test (and then one could in fact simply calculate the spectrum leading to a somewhat circular argument). It would therefore be important to answer the question of what happens if we naively initialise in a different state that is uninformed by the Hamiltonian, e.g., a state that is localised either on one or the other sublattice on one site. Is it possible to make any general statements about self acceleration in such a general case? Is it still a correspondence or

are there cases where self acceleration can occur without non-trivial topology and vice versa (see also point 2)? I believe that it is very important to answer these questions to justify the claim of the abstract (that self acceleration is a “sensitive probe for phenomena in non-Hermitian systems that originate from spectral topology”).

We thank the Referee for raising this important question. The answer to the question is basically rooted in theorems 1 and 2 stated in our previous reply to the Referee. Basically: 1) For the most paradigmatic class of NH lattice models featuring a single band (this case is the one considered for example in the recent paper [2] mentioned by the Referee), the self-acceleration probing method simply requires single-site excitation, so we do not need to have any knowledge of the specific form of the Hamiltonian nor any symmetry of the system. According to theorem 1, under the single-site excitation, self-acceleration is observed if and only if the Bloch Hamiltonian $H(k)$ has a non-trivial point gap topology, i.e., the PBC energy spectrum manifests as a closed loop \mathcal{C} . Additionally, the value of the acceleration provides a measure of the area enclosed by the loop \mathcal{C} . Therefore our method does not have any practical limitations.

What happens in a single-band lattice if we do not excite the system in a single site? For a Hermitian system, self-acceleration always vanishes for an arbitrary initial excitation of the lattice (as shown in the Methods section of our manuscript and according to the Ehrenfest theorem). For a NH system, self-acceleration could be observed also in systems with a trivial point gap topology (the PBC energy spectrum of $H(k)$ describes an open arc).

2) For multi-band systems, indeed according to theorem 2 and Referee’s comment, we require a special excitation of the lattice over several sites in different sublattices [see Eq. (R5)], and rather generally this cannot be done unless we know *a priori* the Hamiltonian of the system. Nevertheless, for narrow-band systems spaced by wide gaps in the complex energy plane, the required initial excitation takes a simple form, and in the appropriate basis it basically reduces to single-site excitation in one of the sublattice sites of the unit cell. Similar to the single-band case, we then do not need a precise knowledge of the Hamiltonian. Therefore, our method can be applied to multi-band lattice systems as well with wide gaps, without any practical limitations or much information about the system Hamiltonian.

In our experiment, we are dealing with a two-band system, however, as discussed in the

Methods section of our manuscript, for a coin angle θ close to $\pi/2$ (which is the narrow-band regime), the corresponding initial excitation is very simple and we can accurately estimate the spectral areas from wave-packet self-acceleration measurements.

To further clarify such a point, let us consider as an example the two-band lattice model of Fig. R2 (the second counter example suggested by the Referee). In Fig. R3, we show some typical initial-excitation conditions satisfying the requirements of theorem 2 [Eq.(R5)] for the ‘‘Hatano-Nelson’’ band (i.e. the band that would reduce to the model (R26) in the large dissipative regime, depicted by the blue curve in the subplots), as the dissipation γ' in the sublattice B is decreased. Note that, when the two PBC energy bands in the complex plane

FIG. R3. The PBC energy spectra of the two-band NH lattice model of Fig. R2(a) for a few decreasing values of the dissipation rate γ' (upper panels) and the corresponding occupation probabilities in the two sublattices A and B of the initial state defined by Eq. (R5) (lower panels). The values of γ' are 8, 4, 1.5 and 1 from (a) to (d), respectively. Other parameters are the same as those in Fig. R2. Note that when the two lattice bands are widely separated [as in panels (a) and (b)] the initial-state excitation of the system is well-approximated by a single-site excitation of the sublattice A.

are widely spaced, as in panels (a) and (b) of Fig. R3, the initial excitation of the lattice basically corresponds to a single-site excitation of sublattice A.

3) Context: In a series of works, a one-to-one correspondence was proven and demonstrated that allows to infer the spectral topology from a straightforward scattering experiment or the steady state. The phenomenon of directional amplification has been linked to spectral non-Hermitian topology [R1-R3] and has been

successfully demonstrated in experiments [R4]. This connection is very strong: indeed, there is a one-to-one correspondence between non-trivial non-Hermitian topology and directional amplification. Performing a simple scattering experiment is sufficient to determine whether the underlying system is non-trivial or not and extract the winding number by counting the number of amplifying channels [R1-R3]. This approach also does not require post-selection and the phenomenon is robust against disorder [R2]. The authors may wish to mention at least [R1] and [R4] to put their work into a wider context.

[R1] Nat Commun 11, 3149 (2020).

[R2] Phys. Rev. Lett. 127, 213601 (2021).

[R3] SciPost Phys. 15, 173 (2023).

[R4] arXiv:2309.05825 (2023).

The effects described in this manuscript and in refs. [R1-R4] may to some extent be connected. In particular, some aspects mentioned by the authors (such as the conditions directional propagation and the fact that the sign of the winding number determines the direction of propagation) are also discussed in the works above.

Furthermore, the authors may find the intuitive explanation of directional amplification as a result of the accumulation of excitations at the system edge (see Figure 2 in [R1]) helpful as they themselves make a related remark. Note that directional amplification does not occur in the counter example mentioned under point 2 (only non-reciprocity), so it accurately predicts the topology also in that case [R3].

We thank the Referee for the helpful advice. We have expanded upon the interconnections between our findings and those presented in references [R1-R4] in the latest version of our manuscript. We have placed particular emphasis on shared features, such as the directional propagation, and the pivotal role played by the winding number in determining the propagation direction. It is crucial to highlight, however, that our new method diverges from the approaches outlined in these references because it offers not only insights into the system's topology, but also it provides information on the *spectral geometry*, specifically the

area of the Hamiltonian spectrum. Notably, our approach distinguishes itself by circumventing the necessity for a scattering setup (as in the driven-dissipative systems mentioned by the Referee). Moreover, in our scheme, the probing occurs within an exceptionally brief time interval, rendering it robust against decoherence and dephasing effects and making post selection not necessary. We have incorporated the useful Referee’s suggestions into the revised manuscript. We also included citations to references [R1-R4] to provide a more comprehensive context for our work.

4) I am a bit confused by role of dynamical stability. Clearly, the dynamics described here will not be dynamically stable (Figs. 1c and 2h) since some eigenstates correspond to eigenvalues with positive imaginary part. What are the consequences on the dynamics in the experiment when the spectrum encircles the origin and some states correspond to unstable modes? Does this imply the experiment can only be performed for a certain time duration (until the amplitudes are too large)? Do physical non-linear effects become relevant?

The lack of dynamic stability is an inherent characteristic of our system. In our experiment, we conduct simulations using a non-unitary Floquet operator U , with subsequent measurements performed by tracking the leaked photons (reflected by the beam splitter) after each time step. This method of measurement allows us to observe the dynamic information of photons at early time steps, which is crucial for the probing of self-acceleration (see our replies above). In our experiments, we mainly focus on the results of the dynamics within 1 – 24 time steps, and the evolved state is always stable within the time step of the detection stage. Furthermore, we have implemented measures to minimize potential nonlinear effects in our experiment. Specifically, we have maintained strict control over the average photon number per pulse, ensuring it remains below 2.6×10^{-4} . This fine control, achieved through fine-tuning of our experimental setup (using the neutral density filter after the input pulse), serves to eliminate the probability of multi-photon events and consequently makes nonlinear effects negligible in our experiment.

5) Presentation: It would improve the accessibility of the manuscript for a wider audience if the authors would elaborate in a short paragraph on the connection between the implemented lattice model and the photonic quantum walk, i.e., the

connection between Eq.(1) and a NH Hamiltonian. In particular, it would help the reader if the authors could explicitly state which lattice model is described by Eq.(1) and after equation (7) since it is not straightforward to infer from these equations and actually never explicitly stated. It would also be helpful to write H_k above Eq.(10). While there is more information about this in the Supplementary, it would improve the presentation if more details were given in the main text surrounding Eq.(1) and a reference to the Supplemental Material was given where due.

Thank you for the above suggestions. In the revised version of the manuscript, we explicitly state which lattice model is described by Eq. (1) and Eq. (7). We cite the Supplemental Material below Eq. (1) and above Eq. (10).

6) Exposition of the experiment: Authors refer to Refs. [53-55] for the experimental techniques to implement a photonic quantum walk claiming that this is a well-established technique (although all cited papers and pre-prints date from 2022). However, given the broad nature of the journal it would help the readers if the authors could give a brief description of their setup (e.g. in the Methods section) and refer to it at appropriate places in the main text. The authors do describe their experimental setup in the Supplemental Material, however, this is currently left for the reader to discover so it would help the presentation if the Supplemental Material was referenced in the main text where appropriate.

Thanks for the suggestions. In the revised manuscript version, we give a brief description of our experimental setup in the Methods section and refer to it above Eq. (1) in the main text.

7) When self acceleration is discussed it may be helpful to give an intuitive explanation for where the effect could come from. My intuition would be that due to the spectral topology some portions of the wave packet are damped more strongly than others resulting in the deformation of the wave packet which on the level of the centre-of-mass motion results in an acceleration. Perhaps the authors have another intuitive picture in mind. A possible place for such a discussion could be where self acceleration is first discussed in the Results section and/ or when the relation to the Ehrenfest theorem is discussed.

Thanks for the advice. Your intuition is correct, the evolution of the wave function in the presence of spectral topology induces a deformation of the wave packet, ultimately leading to the center-of-mass acceleration of the wave packet. The key mechanism here is the Non-Hermitian Skin Effect (NHSE), which inherently imparts self-acceleration to the wave packet. Let us consider a single-band system and single-site initial excitation of the lattice at $t = 0$. Clearly, the initial mean speed of the wave packet is zero, as all momenta k within the Brillouin zone are equally excited. Owing to the non-Hermitian skin effect, Bloch modes displaying opposite group velocities are differently damped or amplified, and their unbalanced interference yields a deformation of the wave packet spreading and a directed transport at long times, characterized by a non-vanishing drift velocity v_m . This implies that the wave packet must necessarily undergo acceleration to attain a final non-zero speed. The resulting self acceleration can be thus explained as an unbalanced interference effects between spectral wave packet components displaying opposite group velocities. As the Referee suggested, we added a brief discussion in the Methods section.

8) Connected to this: how does the shape of the wave packet change during the time evolution? Since the initial state is highly delocalised in k , I would expect strong wave-packet spreading. It would be my expectation that this spreading may after some time even make it difficult to identify the centre of mass unambiguously (for some system parameters which result in fast spreading).

The Referee is correct about the changing shape of the wave packet during the time evolution, which is dependent on the system parameters. For example, in our one-dimension model, the wave packet undergoes fast spreading when we choose the coin parameter $\theta \approx \pi$. However, we emphasize that, despite the fast-spreading observed at $\theta \approx \pi$, measuring the center of mass does not pose any experimental challenges. The reason is that the discretization of lattice sites in our model, coupled with our experimental setup with a time step t up to 24, ensures that there are at most 49 lattice sites. This discretization, along with the limited number of lattice sites, effectively mitigate experimental challenges associated with identifying the center of mass.

9A) Some typos/ grammar improvements/ other comments:

Abstract:

- “Dynamic behaviours [] originates” → “The dynamic behaviour [] originate”
- “on the complex plane” → “in the complex plane”
- “the concomitant dynamics is” → “the concomitant dynamics are”
- “We then reveal similar correspondence” → “We then reveal a similar correspondence”

Introduction:

- “The dynamics of a system is intimately connected” → “The dynamics of a system are intimately connected”
- “Just as the energy of a celestial body impacts its trajectory [5], so the energy quantization accounts for the spontaneous collapse” → “Just as the energy of a celestial body impacts its trajectory [5], the energy quantization accounts for the spontaneous collapse”
- “on the complex plane → “in the complex plane”
- “with non-trivial consequences in the system → “with non-trivial consequences on the system”
- It would be helpful to cite a suitable subset of the following works together with [13-17], e.g. at the very least [R1] and [R4]:

[R1] Nat Commun 11, 3149 (2020).

[R2] Phys. Rev. Lett. 127, 213601 (2021).

[R3] SciPost Phys. 15, 173 (2023).

[R4] arXiv:2309.05825 (2023).

We thank the Referee for his/her careful reading of our manuscript, we corrected all the typos and cited new references as suggested.

9B) Results:

- Above Eq. (2): “Apparently, the quantum walk implements a stroboscopic simulation of the Hamiltonian H ” → “The quantum walk implements a strobo-

scopic simulation of the Hamiltonian H [ref]”. Here, a citation and/ or reference to the Supplementary Material or Methods where this is shown would be helpful.

We cited a reference to the Supplementary Material as suggested by the Referee.

- Below Eq. (4): “with θ being quite close to $\pi/2$ ” \rightarrow it would be helpful to be more precise here.

We now provide a more precise parameter, i.e., $\theta = 0.45\pi$.

- The authors state that the eigenvalues given in Eq. (5) resemble that of a Hatano-Nelson model. However, the model they implement seems to have some sub-lattice structure so it would be helpful to precisely state which model is actually implemented (i.e., give either the lattice Hamiltonian or H_k). - Below Eq. (5): “The experimental implementation of such a local initial state $|\square\rangle$ is discussed in the Supplemental Material” \rightarrow it would be helpful if more information about the experiment (perhaps just a brief exposition would be given in the main text or the Methods section.

We now give a lattice Hamiltonian below Eq. (5) and give a brief exposition of the experimental implementation for the local initial state in the Method section.

- Below Eq. (7): specify the lattice model and / or H_k that is implemented.

We now provide a specific lattice model below Eq. (7).

- Below Eq. (7): “directional propagation in the two-dimensional plane” \rightarrow Along x or y or both?

We now made it clear that both x and y are involved in the directional propagation.

- Eq. (8): k_mx and k_my should be explicitly defined.

We now state clearly that $E_+(k_x = k_{mx}, k_y = k_{my})$ corresponds to the largest imaginary part of $E_+(k_x, k_y)$.

Methods: - Eq. (9) and (10): it would be helpful to write H_k explicitly.

The precise formulation of the Hamiltonian is complicated, but we refer to its explicit form in the Supplemental Material.

- $|\Psi(k)\rangle$ in Eqs. (11), (13), and (15) are unnormalised. Since the normalisation constant in (15) will depend on k and the integral is taken over k , this will be important. The authors should check that their conclusions are correct once they include the k -dependent normalisation constant. The missing normalisation constant may also have an impact on Eq. (18).

We check that the results are correct since the normalization constant depends on γ only, which can be eliminated in Eq. (16).

- Eq. (14) and (15): for completeness, it should be mentioned that at finite system size the sum over k is approximated by integral.

We added the description below Eq. (14).

- Above Eq. (19): it would be helpful to refer to the equation where $\tilde{E}(k_x, k_y)$ is defined.

$\tilde{E}(k_x, k_y)$ is defined in Eq. (19).

- Page 6, right column: “Universality of self acceleration” is a strong statement given the restrictions of the correspondence already remarked on by the authors.

The statement here only points toward the one-dimensional system and aims to explain the origin of self-acceleration in non-Hermitian systems. To avoid misunderstanding, we have changed the statement to: “Origin of self acceleration ”

- Above Eq. (27): “the evolved wave function in the early times” \rightarrow “the evolved wave function for early times”

We corrected it.

- Eq. (27): time dependent normalisation constant is missing.

The theoretical analysis remains unaffected by the time-dependent normalization constant since it can be eliminated by the definition of the mean position $\langle x \rangle = \langle \psi_t | x | \psi_t \rangle / \langle \psi_t | \psi_t \rangle$.

Reply to Report of Referee #2

After carefully reading the work by P. Xue et al., entitled “Self acceleration from spectral geometry in dissipative quantum-walk dynamics,” I regret to inform that I cannot recommend it for in Nature Communications.

We thank the Referee for carefully reviewing our manuscript. It is regrettable that the Referee cannot approve the publication of our work.

My main concern is that the manuscript does not satisfy the publication criteria for Nature Communications. Specifically, the Aims & Scope state that Papers published by the journal aim to represent important advances of significance to specialists within each field. Although the manuscript is well written, and the results are interesting; unfortunately, it does not provide significant advances, either theoretical or experimental, to the field which it belongs. To be more specific, I found a tremendous overlap between the theory presented in Ref. [51]: Phys. Rev. B 105, 245143 (2022), and the manuscript under review. Please note that the correspondence between self-acceleration and spectral topology is already discussed in the 2022 paper. On the other hand, the experimental setup used in the manuscript under review has been previously introduced in Nat. Commun. 14, 6283 (2023), meaning that new experimental tools/techniques are not demonstrated either.

From the discussion above, although I believe the experimental verification of the theoretical results predicted in Ref. [51] deserve to be published; it is my considered opinion that Nature Communications is not the proper journal.

We appreciate your positive remarks regarding the interesting nature of our results and the quality of our writing. We would like to comprehensively address your concerns from both theoretical and experimental standpoints.

First, the theoretical paper of Ref. [51] establishes the one-to-one correspondence between non-trivial spectral point gap topology and self-acceleration, in the form of the following theorem: *In any single-band NH lattice model, the early-time acceleration a_x of the center-of-mass wave packet under initial single-site excitation of the lattice is proportional to the*

spectral area \mathcal{A} enclosed by the complex PBC energy spectrum $H(k)$ in the complex plane, namely

$$a_x = \frac{2}{\pi} \mathcal{A}. \quad (\text{R27})$$

And our work is the *first* experiment on any physical platform to confirm such a main theoretical result connecting spectral geometry of a non-Hermitian system with measurable dynamical variables. We would like to emphasize here that knowing something theoretically by no means implies that proving it experimentally is obvious or trivial in physics (and in natural sciences in general). On the contrary, an experimental test of an interesting theoretical idea has been a major driving force for advances in physics and is therefore not only necessary but also scientifically vital. To this regards, photonics and ultracold atoms in optical lattices have provided leading and experimentally accessible laboratory tools to emulate synthetic matter and to observe major physical phenomena in condensed-matter physics theoretically known since long time but hardly observable in the electronic systems, such as Anderson localization, Bloch oscillations, quantum chaos, Floquet topological insulators, anomalous Floquet topological insulators, Anderson insulators etc. to mention a few. The first experimental demonstrations in photonics of such theoretically well-known phenomena, published in high-impact journals (such as *Nature* and *Science*), have provided a main driving force in research opening up new areas of investigations, such as disordered and topological photonics. We believe that our first experimental observation of spectral geometry in non-Hermitian systems from dynamical measurements, which can avoid detrimental effects of decoherence in the system, provides a main experimental advance in the rapidly emerging area of non-Hermitian topological physics, and thus worth for publication in *Nature Communications*.

Even more importantly, our work goes beyond the mere observation and confirmation of the theorem above, by studying also the *multi-band* case, as well as the self-acceleration in two-dimensional models. These two aspects are not discussed by any theoretical works prior to our experiment. As a matter of fact, our experiments on quantum walks on synthetic lattice inherently concern a *multiband* (two-band) system, and previous analysis of Ref.[51] cannot be applied and requires some nontrivial extensions of the theory.

To remark such a major and novel point, in the revised version of our manuscript we present new theoretical analysis considering self-acceleration in multi-band models, and experimentally confirmed our theoretical prediction. Namely, we state and prove the following

theorem (see theorem 2 in the revised Supplemental Material): *Let us consider coherent wave dynamics in a multiband NH lattice. For a given band index α of the lattice ($\alpha = 1, 2, \dots, M$), let us prepare the system at the initial time $t = 0$ in the state*

$$a_n(0) = \frac{1}{2\pi} \int_{-\pi}^{\pi} A_\alpha(k) \exp(ikn) , b_n(0) = \frac{1}{2\pi} \int_{-\pi}^{\pi} B_\alpha(k) \exp(ikn) , c_n(0) = \frac{1}{2\pi} \int_{-\pi}^{\pi} C_\alpha(k) \exp(ikn) \dots$$

where the Bloch eigenvector amplitudes $A_\alpha(k)$, $B_\alpha(k)$, ... are assumed to satisfy Eq. (R3). Then, if the system has a trivial energy-surface topology, the early-time acceleration a_x of the wave packet center of mass is given by

$$a_x = \frac{2}{\pi} \mathcal{A}_\alpha,$$

where \mathcal{A}_α is the area enclosed by the curve $E_\alpha(k)$ in the complex plane.

We remark that to demonstrate in the experiment the correspondence between self-acceleration and spectral geometry, we require special care in the preparation of the system and composite quantum walk runs are necessary, as discussed in the manuscript. Furthermore, based on the flexible control and feasible degrees of freedom in photonic quantum walks, we extended our setup to two-dimensional systems, and discuss (both in theory and using experiments) the fate and condition for the correspondence in two dimensions.

From an experimental perspective and in the domain of dissipative (and hence gainless) discrete-time quantum walks, our experiment here stands out as achieving the largest number of evolution steps in both one- and two-dimensions. This is a highly non-trivial achievement, considering the significant time investment required for fine-tuning optical components and enhancing experimental system stability. Such a dedicated effort is necessary, since the accurate portrayal of the system's dynamics necessitates a long evolution time and larger quantum-walk steps. In our experiment, a limited number of simulation steps proves to be insufficient to unveil the transition between the short-time self-acceleration dynamics and the long-time directional flow with uniform drift velocity. This is clearly visible in Fig. 5(a) of the main text, where a 16-step evolution [the largest number of time steps in Nat. Commun. 14, 6283 (2023)] is not sufficient. Therefore, also from a technical viewpoint our current experiments provide significant progress, pushing the number of measurable time steps in a passive system far beyond previous achievements.

To summarize, we believe that the novel theoretical insights and experimental achievements presented in the revised version contribute valuable advancements to the field, and meet the high standards of Nature Communications.

Minor comment: Although using the term "self-acceleration" might sound attractive, I believe the authors should explain the reader that this is not a true accelerating optical signal, but an interference pattern that results from the influence of the non-Hermiticity on the lattice modes. It is interesting to note from Figure 2 that the effect of the "gain-loss" parameter seems to be to unbalance the coin, thus making the quantum walk to move toward one specific direction. I wonder if changing the signs of γ in Fig. 2b and 2c makes the slopes of the plots change their direction as well.

We thank the Referee for this interesting comment and remark. We fully agree that the self acceleration phenomenon arises from an interference pattern of Bloch modes that is influenced by the non-Hermitian skin effect in the lattice. Clearly, we do not have any optical signal accelerating on the lattice, rather a wave packet resulting from the delicate interference of all Bloch modes excited at initial time in the lattice. Assuming a single-band system and single-site initial excitation of the lattice at $t = 0$, the initial mean speed of the wave packet is clearly zero, as all momenta k within the Brillouin zone are equally excited. Owing to the non-Hermitian skin effect, Bloch modes displaying opposite group velocities are differently damped or amplified, and their unbalanced interference yields a deformation of the wave packet spreading and a directed transport at long times, characterized by a non-vanishing drift velocity v_m . This implies that the wave packet must necessarily undergo acceleration to attain a final non-zero speed. The resulting self acceleration can be thus explained as an unbalanced interference effect between spectral wave packet components displaying opposite group velocities. In the revised manuscript, method section, we have now clarified the physical wavy interference effect underlying the appearance of wave packet acceleration on the lattice.

Finally, as correctly stated by the Referee, reversing the sign of γ corresponds to a change of the direction of the wave packet motion.

Reply to Report of Referee #3

In the work titled “Self acceleration from spectral geometry in dissipative quantum-walk dynamics”, the Authors demonstrate, theoretically and experimentally a short-time acceleration phenomenon associated with the spectral features of a non-hermitian process. In non-hermitian systems, the Hamiltonian has complex eigenvalues. This work shows that the dispersion relations of the energy’s real and imaginary part are responsible for a parabolic trajectory of the center of mass (when considering the short time dynamics of initial states that fill uniformly a given band). This effect is clearly demonstrated in 1D and 2D quantum walks performed in the time bin degree of freedom. Both the theoretical and experimental results sound, novel (to the best of my knowledge) and can unveil new strategies to understand the physics of non-hermitian systems.

I thus recommend this work for publication in Nature Communications.

We thank the Referee for his/her careful reading of our manuscript and for recommending our work for publication in Nature Communications.

Some minor comments,

- The authors may expand on the possible applications of their result. For example, they may discuss the possibility of monitoring self acceleration to probe some specific features of the system: is there any information about the topological phase that can be inferred? If initial states are prepared occupying more bands, are there possible interferometric protocol that can be devised, like in Landau-Zener-Stucklerberg interferometry?

We appreciate the insightful Referee’s comments and suggestions regarding the possible applications of our results. As we elaborate in the manuscript, the measured short-time acceleration of the wave function is proportional to the area enclosed by the eigenspectrum. Combined with the correspondence between the spectral winding number and the sign of the acceleration, we are able to probe not only the topological but also the geometric features of the eigenspectrum in the complex plane. However, the study here is not related to the band topology (or the relevant topological phases), which requires the probing of other dynamic quantities such as the chiral displacement [c.f. Phys. Rev. Lett. 127, 270602 (2021)].

In the context of initial states occupying multiple bands, self-acceleration could still be observed in systems with point-gap topology under some conditions, as we discuss in details in a new Section of the Supplementary material. In the revised version of our manuscript, we expand the theoretical analysis to multi-band systems and reveal the conditions to get a one-to-one correspondence between self-acceleration and spectral areas. Further, while an interferometric protocol can indeed be developed, making use of the short-time self-acceleration dynamics reported here, we decide to focus on the fundamental observation of the phenomena as well as the confirmation of its extensions in multi-band and two-dimensional systems. For future studies, it is indeed of interest to consider this possibility, particularly following the example of Landau-Zener-Stücklerberg interferometry, given that Zener tunneling in the presence of the skin effect can display significant chiral behavior [c.f. Phys. Rev. Lett. 124, 066602 (2020)].

-
- [1] N. Okuma, *et al.*, Topological origin of non-Hermitian skin effects. *Phys. Rev. Lett.* **124**, 086801 (2020).
 - [2] M. Brunelli *et al.*, Restoration of the non-Hermitian bulk-boundary correspondence via topological amplification. *SciPost Phys.* **15**, 173 (2023).
 - [3] S. Longhi, Non-Hermitian skin effect and self-acceleration. *Phys. Rev. B* **105**, 245143 (2022).
 - [4] W.D. Heiss, The physics of exceptional points. *J. Phys. A* **45**, 444016 (2012).
 - [5] S. Longhi, Floquet exceptional points and chirality in non-Hermitian Hamiltonians, *J. Phys. A: Math. Theor.* **50**, 505201 (2017).
 - [6] M.-A. Miri, *et al.*, Exceptional points in optics and photonics. *Science* **363**, eaar7709 (2019).
 - [7] S. Longhi, *et al.*, Complex Berry phase and imperfect non-Hermitian phase transitions. *Phys. Rev. B* **107**, 085122 (2023).
 - [8] C.C. Wanjura, *et al.*, Topological framework for directional amplification in driven-dissipative cavity arrays. *Nat. Commun.* **11**, 3149 (2020).

List of changes

1. We added the theorem for multi-band lattice models in the revised Supplemental Material and clarified the one-to-one correspondence between self acceleration and spectral area.
2. As recommended by Referee 1, we included the additional references in the revised main text.
3. As recommended by Referee 1, we added citations to the Supplemental Material and Methods in the revised main text.
4. We added a brief description of our experimental setup in the revised Methods section.
5. We corrected every typo and accepted every grammar improvement pointed out by Referee 1.
6. We provided a more precise parameter about θ below Eq.(4) in the revised main text.
7. We provided the lattice Hamiltonian below Eq. (5) in the revised main text.
8. We added a brief exposition of the experimental implementation for the initial state in the revised Method section.
9. We provided the specific lattice model below Eq. (7) in the revised main text.
10. We clarified that the directed propagation below Eq.(7) involves both x and y in the revised main text.
11. The definition of $E_+(k_x = k_{mx}, k_y = k_{my})$ has now been made explicit in the revised Methods section.
12. We added the description below Eq. (15) stating that the integral approximates the sum over k at finite system size in the revised Methods section.
13. We changed the statement from “Universality of self acceleration” to “Origin of self acceleration” in the revised Method section.
14. We provided in the Method section a physical explanation of the origin of the self acceleration.

REVIEWER COMMENTS

Reviewer #1 (Remarks to the Author):

The authors have made a serious attempt at responding to all my concerns and added a substantial amount of new material to the supplementary information. While I appreciate their effort and they resolved some of my comments, one important concern remains unresolved and I believe that some of the new statements (as well as Fig. R2) are false. Specifically, they write the following in the reply:

As Fig. R2 clearly shows, we still have a point gap topology even when we do not eliminate the auxiliary modes from the analysis, and, according to [1], we still have the NH skin effect. Therefore, the Referee's statement that the NH skin should disappear, when the auxiliary modes are integrated out, seems from our analysis an incorrect statement. For this model, we have directly checked the persistence of the NH skin effect in all cases (even when the energy-surface topology is nontrivial) by numerical computation of the Hamiltonian eigenstates in a lattice with open boundaries comprising $N = 100$ unit cells; our results clearly show a preferential localization to one edge of the lattice.

This statement is false which I have explicitly double-checked numerically (see below for some plots). The model in which the auxiliary modes have not been integrated out (from here on 'the full model') *does not display the NH skin effect and the point gap remains closed.*

For the case $\gamma = \gamma'$ (but this is also true in the general case), this is straightforward to understand: that model is, as the authors themselves state, essentially a Hermitian model with added uniform local dissipation gamma, i.e. $H_{\text{NH}} = H_{\text{Hermitian}} - i\gamma I$ with $H_{\text{Hermitian}}$ the Hermitian part of the NH Hamiltonian. In this case, the spectrum should always be of the form $E = E_{\text{real}} - i\gamma$ with E_{real} the real spectrum of the Hermitian model. Similarly, the eigenvectors should be those of the Hermitian model, so this full model cannot display the NH skin effect. Only after tracing out the auxiliary modes, the NH skin effect appears and the PBC spectrum opens a point gap.

To support my statement, I show below some plots of the eigenvalues and eigenvectors before integrating out the auxiliary modes. After integrating out the auxiliary modes, the PBC spectrum will display a point gap and the OBC eigenvectors localise (NH skin effect).

Spectrum and eigenvectors before integrating out the auxiliary modes:

(The first 50 sites are the system modes, the next 49 sites are the bath auxiliary modes and the next 50 modes are the modes that provide gain to the system via two-mode-squeezing (the authors did not include those modes in their model, but this does not change the qualitative result.)

OBC:

PBC:

As expected for a normal matrix (for $\gamma = \gamma'$, the matrix in (R25) is normal), the PBC and OBC spectra agree in the full models and clearly the OBC eigenvectors do not localise.

The reason why the spectrum of the full system and the reduced system are so different is because the connection between them is established via a non-unitary map (which does not preserve the spectrum).

As stated in my previous report, if the dynamics of the full system also exhibit self-acceleration (as the authors claim, although I did not double-check this), the above would be a counter-example in which self-acceleration is *not* in one-to-one correspondence with open point-gap topology (as in the above example self-acceleration can also occur *without* open point-gap topology). Undoubtedly, there is a strong connection between self-acceleration and point gap topology, but I suspect that one has to be more precise in stating when it holds.

It is unclear to me what lead the authors to making this mistake in their analysis and the numerics. However, this issue (double-checking their numerics and clarifying if it is really a one-to-one correspondence) has to be resolved as it is important to the main claim of their paper.

Reviewer #2 (Remarks to the Author):

I would like to thank the authors for the effort in preparing a complete response letter addressing all referees' concerns.

After reading the revised manuscript and the clarifications regarding the new materials presented in the paper, I am inclined to recommend it for publication.

My main motivation for the above lies in the description (provided in the response letter) of the new materials regarding the extension of the work published in Ref. [55] of the revised manuscript, namely the two-dimensional and multiband models, along with their experimental demonstration. As I mentioned in my previous report, these results are interesting and deserve to be published, but the reason as to why they deserve publication in a high-impact journal, such as Nature Communications, was not entirely clear to me.

I have only one point, in which I do not agree with the authors:

"This is a highly non-trivial achievement, considering the significant time investment required for fine-tuning optical components and enhancing experimental system stability."

Being an experiment-oriented scientist myself, I can say that any experiment requires time investment and its difficulty in no way defines its importance in a research field/community or journal. It is the impact of the work that makes it worth sharing it with the community.

**Response to the review reports of the manuscript NCOMMS-23-46566A “Self
acceleration from spectral geometry in dissipative quantum-walk dynamics”
(SECOND ROUND)**

Reply to Report of Reviewer #1

The authors have made a serious attempt at responding to all my concerns and added a substantial amount of new material to the supplementary information. While I appreciate their effort and they resolved some of my comments, one important concern remains unresolved and I believe that some of the new statements (as well as Fig. R2) are false.

We thank the reviewer for appreciating our efforts in adding a substantial amount of new material and solving some of the concerns raised in his/her previous report. However, the reviewer raises another concern by displaying some numerical results, which seem to contradict our conclusions on the general validity of the correspondence between self-acceleration and NH skin effect. As detailed below, we believe the remaining concern of the reviewer stems from a misunderstanding. In fact, there are no contradictions nor flaws in our analysis. We can reproduce the reviewer’s results, and we can explain how to conciliate our statements and results with those found by the reviewer, thus overcoming the reviewer’s concern and re-instating the general validity of our conclusions, which are rigorously proven in Theorems 1 and 2 for single-band and multi-band non-Hermitian lattices.

Specifically, they write the following in the reply:

As Fig. R2 clearly shows, we still have a point gap topology even when we do not eliminate the auxiliary modes from the analysis, and, according to [1], we still have the NH skin effect. Therefore, the reviewer’s statement that the NH skin should disappear, when the auxiliary modes are integrated out, seems from our analysis an incorrect statement. For this model, we have directly checked the persistence of the NH skin effect in all cases (even when the energy-surface topology is nontrivial) by numerical computation of the Hamiltonian eigenstates in a lattice with open boundaries comprising $N = 100$ unit cells; our results clearly show a preferential localization to one edge of the lattice.

This statement is false which I have explicitly double-checked numerically (see

below for some plots). The model in which the auxiliary modes have not been integrated out (from here on 'the full model') does not display the NH skin effect and the point gap remains closed. For the case $\gamma = \gamma'$ (but this is also true in the general case), this is straightforward to understand: that model is, as the authors themselves state, essentially a Hermitian model with added uniform local dissipation γ , i.e. $H_{NH} = H_{Hermitian} - i\gamma I$ with the Hermitian part of the NH Hamiltonian. In this case, the spectrum should always be of the form $E = E_{real} - i\gamma$ with E_{real} the real spectrum of the Hermitian model. Similarly, the eigenvectors should be those of the Hermitian model, so this full model cannot display the NH skin effect. Only after tracing out the auxiliary modes, the NH skin effect appears and the PBC spectrum opens a point gap. To support my statement, I show below some plots of the eigenvalues and eigenvectors before integrating out the auxiliary modes.

We thank the reviewer for raising this important concern, which we have examined carefully. We fully agree with the reviewer's results and argumentation: when $\gamma = \gamma'$, the Hamiltonian of the full system takes the form $H_{NH} = H_{Hermitian} - i\gamma I$; the energy spectrum is not point-gapped; it differs from that of the Hermitian matrix $H_{Hermitian}$ by a shift $-i\gamma$ in the complex plane; and we do not have the NH skin effect. We can of course reproduce such a result.

However, this is a very special case. For $\gamma \neq \gamma'$, the energy spectrum becomes point-gapped and the skin modes appear under OBC indeed (while the reviewer seems to believe that the disappearance of the skin effect is also observed for $\gamma \neq \gamma'$). We believe that the reviewer found the absence of the NH skin effect just because he/she runs the simulations under the very special condition $\gamma' = \gamma$, without checking the more general cases of $\gamma \neq \gamma'$. Most importantly, when we operate at the special case $\gamma' = \gamma$, according to Theorem 2 (the theorem is given in the Supplementary Material), we *do not* observe any acceleration once the initial state is tailored according to Eq.(S10) of Theorem 2.

To summarize, we fully agree with the reviewer that for $\gamma = \gamma'$ the full model does not display the NH skin effect, but in this case the acceleration also vanishes (we have confirmed this numerically, as shown below). On the other hand, when $\gamma \neq \gamma'$, the full model displays the NH skin effect, and correspondingly we observe a non-vanishing acceleration, which

increases with the area subtended by the spectral loop on the complex plane under the PBC. We believe this should fully resolve the reviewer’s concern.

FIG. R1. Schematic of the full model described by Eqs. (R1) and (R2).

Let us provide some results and clarifications that support the above statements.

We consider, as “full model”, the binary lattice system shown in Fig. R1. As in our previous reply, we do not consider the auxiliary sites that yield gain (see note [1]). In the physical space, the time-dependent Schrödinger equation of the full model reads

$$i \frac{da_n}{dt} = -i\gamma a_n + J \exp(i\phi) a_{n+1} + J \exp(-i\phi) a_{n-1} + J'(b_n + b_{n-1}), \quad (\text{R1})$$

$$i \frac{db_n}{dt} = J'(a_n + a_{n+1}) - i\gamma' b_n, \quad (\text{R2})$$

where $J \exp(\pm i\phi)$ are the left/right hopping amplitudes between adjacent sites in the main sublattice A with effective on-site loss rate γ , J' is the hopping rate with the auxiliary sites of sublattice B with the loss rate γ' , and ϕ is the Aharonov-Bohm phase. This is a two-band system with the NH Bloch Hamiltonian

$$H_{NH}(k) = \begin{pmatrix} -i\gamma + 2J \cos(k + \phi) & J' + J' \exp(-ik) \\ J' + J' \exp(ik) & -i\gamma' \end{pmatrix}. \quad (\text{R3})$$

As correctly stated by the reviewer, in the very special (and accidental) case of $\gamma = \gamma'$, we can write $H_{NH}(k) = H_{\text{Hermitian}}(k) - i\gamma I$ with $H_{\text{Hermitian}}$ being a Hermitian matrix with an entirely real energy spectrum, so that the PBC energy spectrum of $H_{NH}(k)$ is not point-gapped, i.e. it does not describe closed loops in the complex energy plane (it is just the spectrum of $H_{\text{Hermitian}}(k)$ shifted by $-i\gamma$ in the complex plane). Under the OBC, we do

FIG. R2. Left panels: Energy spectra under PBC (solid red curves) and OBC (open black circles) of the NH Hamiltonian of the full model described by the lattice of Fig. R1. In (a) $\gamma' = 1$, in (b) $\gamma' = 1.1$, in (c) $\gamma' = 1.3$. Other parameters are $\gamma = 1$, $J = 0.3$, $J' = 1$ and $\phi = \pi/3$. A lattice comprising $N = 100$ sites in both sublattices A and B has been adopted for the numerical simulations. Right panels: corresponding behavior of 10 representative eigenvectors of the NH Hamiltonian under the OBC. The indices from $n = 1$ to $n = N = 100$ refer to sites of sublattice A, whereas indices from $n = 101$ to $n = 2N = 200$ refer to sites of sublattice B (auxiliary cavities). Note that in (a), corresponding to the special condition $\gamma = \gamma'$, the PBC energy spectrum is not point-gapped and there is no skin effect (this case basically reproduces the reviewer's results). As soon as γ' slightly deviates from γ [panels (b) and (c)], the PBC energy spectrum becomes point-gapped, and the eigenvectors clearly display the skin effect.

not have the NH skin effect, i.e. the bulk eigenmodes are not squeezed toward the lattice edges. This is shown in panel (a) of Fig. R2, which depicts the numerically-computed PBC and OBC energy spectra (which basically coincide) and the amplitude profiles of 10 representative eigenvectors of the NH Hamiltonian under the OBC. We assumed a lattice comprising $N = 100$ sites in both sublattices A and B, and used the parameters $J = 0.3$, $J' = 1$, $\phi = \pi/3$ and $\gamma = \gamma' = 1$. For this special case $\gamma = \gamma'$, we thus recover and confirm the reviewer's results: the energy spectrum is not point-gapped, and there is not the skin effect. The main and key point that directly addresses the reviewer's concern is that, in this special case, **we do not observe self-acceleration** of the wave packet, as shown in Fig. R3 (discussed below), which is consistent with our Theorem 2 on the wave-packet acceleration in a multi-band model.

However, as soon as we take $\gamma' \neq \gamma$, the PBC energy spectrum becomes point-gapped and correspondingly the eigenvectors of the NH Hamiltonian of the full model clearly displays the NH skin effect, i.e. they tend to be squeezed toward one edge of the lattice. Examples of PBC-OBC energy spectra and amplitude profiles of representative eigenvectors of H_{NH} under OBC with $\gamma' \neq \gamma$ are shown in Figs. R2(b) and (c). In such cases, according to Theorem 2 **we do observe acceleration of the wave packet**. This is numerically confirmed in Fig. R3 (see discussions below), and the acceleration is proportional to the area enclosed by the PBC energy spectral loops, consistent with our prediction.

As stated in my previous report, if the dynamics of the full system also exhibit self-acceleration (as the authors claim, although I did not double-check this), the above would be a counter-example in which self-acceleration is not in one-to-one correspondence with open point-gap topology (as in the above example self-acceleration can also occur without open point-gap topology). Undoubtedly, there is a strong connection between self-acceleration and point gap topology, but I suspect that one has to be more precise in stating when it holds.

The full model of Fig. R1, with auxiliary cavities, is a two-band model and thus the relation between the self-acceleration and spectral area is established rigorously in Theorem 2 (in the Supplemental Material). We made further numerical simulations by solving Eqs. (R1) and (R2) in the time domain and confirmed the conclusion. When we initially excite the

lattice in the state defined by Eq. (S10) (as dictated by Theorem 2), we **do not observe** self-acceleration for $\gamma = \gamma'$, where the point gap topology and skin effect are also absent. However, **we do observe** self-acceleration when $\gamma \neq \gamma'$, where we also have point-gap topology and skin effect: the self-acceleration increases as the spectral area enclosed by the PBC energy loop is increased according to Theorem 2. Figure R3 shows the numerically-computed early temporal evolution of the wave-packet center of mass $\langle n(t) \rangle$ under the same parameters as those in Fig. R2. It clearly shows the absence of self-acceleration under the special condition $\gamma = \gamma'$. Therefore, the case $\gamma = \gamma'$ is not a counter-example that violates the one-to-one correspondence between the self-acceleration and spectral area, established by Theorem 2.

FIG. R3. Left panel: Numerically-computed temporal evolution of the wave-packet center of mass $\langle n(t) \rangle$ in the full model for the three cases a, b and c of Fig. R2. The initial excitation of the lattice at $t = 0$, $a_n(0)$ and $b_n(0)$, is tailored according to Eq. (S10) of Theorem 2. Such excitations for the three cases a,b and c are depicted in the right panels. Note that in case a, where the model does not display the NH skin effect, the wave-packet center of mass remains at rest, corresponding to a vanishing self-acceleration.

It is unclear to me what lead the authors to making this mistake in their analysis and the numerics. However, this issue (double-checking their numerics and clarifying if it is really a one-to-one correspondence) has to be resolved as it is important to the main claim of their paper.

We believe that we have fully clarified the reviewer's main concern. There is no conflict of any kind between our numerical results and the reviewer's numerical results, and our conclusion holds as it is: the correspondence between self-acceleration and spectral area, as rigorously stated by Theorem 2 demonstrated in the Supplementary Material, is universally valid and it is not violated in the special case mentioned by the reviewer.

[1] The reviewer does not provide an explicit form of the Hamiltonian of the full system nor parameters used for his/her simulations, so we consider here the model with only a set of auxiliary cavities; this should not be a major issue since the reviewer claims that "the authors did not include those modes in their model, but this does not change the qualitative result". We can of course consider a more complete model if the reviewer provides us with an explicit form of the Hamiltonian he/she would like to have a check.

Reply to Report of Reviewer #2

We are pleased that Reviewer #2 finds the extended materials interesting and satisfactory, and we thank the reviewer for recommending the revised manuscript for publication in Nature Communications. We also agree with the reviewer that it is the scientific impact rather than the time investment and technical difficulties that eventually defines the significance of the work.

REVIEWERS' COMMENTS

Reviewer #1 (Remarks to the Author):

The authors have answered all my questions to my satisfaction and I am now very happy to recommend publication. Especially the last clarification has demonstrated that self-acceleration is an effect that is independent of other (steady state scattering) phenomena such as non-reciprocity (which was my original question) but is really only a consequence of non-trivial point gap topology. As such, I believe this paper makes an important contribution to the field.

Reply to Report of Referee #1

Reviewer #1 (Remarks to the Author):

Comment: “The authors have answered all my questions to my satisfaction and I am now very happy to recommend publication. Especially the last clarification has demonstrated that self-acceleration is an effect that is independent of other (steady state scattering) phenomena such as non-reciprocity (which was my original question) but is really only a consequence of non-trivial point gap topology. As such, I believe this paper makes an important contribution to the field.”

Reply: We thank Reviewer #1 for their time. We are delighted to see that Reviewer #1 is willing to recommend publication of our paper and believes our paper makes an important contribution to the field.